# Loss-cone instability modulation due to a magnetohydrodynamic sausage mode oscillation in the solar corona

Eoin P. Carley[1,2], Laura A. Hayes[1,6], Sophie A. Murray [1,2], Diana E. Morosan[1,3], Warren Shelley [1,7], Nicole Vilmer[4,5] & Peter T. Gallagher [1,2]

Solar flares often involve the acceleration of particles to relativistic energies and the generation of high-intensity bursts of radio emission. In some cases, the radio bursts can show periodic or quasiperiodic intensity pulsations. However, precisely how these pulsations are generated is still subject to debate. Prominent theories employ mechanisms such as periodic magnetic reconnection, magnetohydrodynamic (MHD) oscillations, or some combination of both. Here we report on high-cadence (0.25 s) radio imaging of a 228 MHz radio source pulsating with a period of 2.3 s during a solar flare on 2014-April-18. The pulsating source is due to an MHD sausage mode oscillation periodically triggering electron acceleration in the corona. The periodic electron acceleration results in the modulation of a loss-cone instability, ultimately resulting in pulsating plasma emission. The results show that a complex combination of MHD oscillations and plasma instability modulation can lead to pulsating radio emission in astrophysical environments.

[1] School of Physics, Trinity College Dublin, Dublin 2, Ireland. [2] School of Cosmic Physics, Dublin Institute for Advanced Studies, D02 XF85 Dublin, Ireland. [3] Department of Physics, University of Helsinki, P.O. Box 64 Helsinki, Finland. [4] LESIA, Observatoire de Paris, PSL Research University, CNRS, Sorbonne Université, UPMC Univ. Paris 06, Univ. Paris Diderot, Sorbonne Paris Cité, 5 place Jules Janssen, 92195 Meudon, France. [5] Station de Radioastronomie de Nançay, Observatoire de Paris, PSL Research University, CNRS, Univ. Orléans, 31 Fitzwilliam Place, Nançay 18330, France. [6] Present address: Solar Physics Laboratory, Code 671, Heliophysics Science Division, NASA Goddard Space Flight Center, Greenbelt, Maryland, MD 20771, USA. [7] Present address: Department of Astronomy, Boston University, 725 Commonwealth Ave., Boston, MA 02215, USA. Correspondence and requests for materials should be addressed to E.P.C. (email: eoin.carley@tcd.ie)

Solar flares are the most energetic phenomena in the solar system, thought to be due to the release of up to $10^{25}$ J of magnetic energy over the course of tens of minutes[1]. This energy release results in the acceleration of particles to relativistic energies and the generation of electromagnetic radiation from gamma rays to high-intensity bursts of radio emission[2]. The radio emission at metric wavelengths is thought to be generated by a plasma instability due to the presence of energetic electrons in the solar corona[3], and can sometimes show periodic or quasiperiodic pulsations in intensity[4,5]. Precisely what causes the periodic pulsations is still subject to much investigation. The proposed explanations include periodic magnetic reconnection, intensity modulation via magnetohydrodynamic (MHD) oscillations[5–7], or a cyclic energy exchange between the plasma particle and wave distributions modelled by a Lotka-Volterra system[6]. Direct high-cadence radio imaging of the behaviour of electron beams during a pulsation event has the potential to confirm one (or some combination) of these mechanisms. A confirmation of which mechanism is responsible for solar radio pulsations can also allow for a comparison with periodically induced plasma instabilities in other astrophysical systems[8] and Earth-based laboratories[9–11].

Solar radio pulsations usually display periodicities from seconds to minutes[6] and are observed from metric to millimetric wavelengths[12,13]. They are thought to be associated with a repeated acceleration of electrons during a solar flare or a periodic modulation of the radio-emitting flare plasma[6,14–16]. Although they are rarely imaged at radio wavelengths[16–18], comparison of the phenomenon with extreme ultraviolet (EUV) and X-ray imaging has shown that the locations of such pulsating sources are among concentrations of magnetic loops in the solar atmosphere[19–22], known as active regions. Refs. [16,18,23] have attributed the pulsations to periodic injection of electrons into loop footpoints in the active region, which act as a magnetic trap. This results in modulated maser or plasma emission from a loss-cone instability induced in the trap.

While radio pulsations are thought to be due to the periodic pulsing of the plasma or electron properties in the corona[18,24,25], there is ongoing theoretical debate on the cause of the periodicity. Magnetic reconnection is a primary candidate for such a mechanism, since its dynamics can be either periodic or quasiperiodic in time[26–28]. An alternative candidate is the presence of MHD sausage mode oscillations in magnetic loops, which tend to have short periods of 1–10 s in coronal conditions[12,29,30]. Such oscillations can modulate a loss-cone instability at loop footpoints, which in turn modulates any radio emission that is driven by the instability. A small number of studies have employed a hybrid of the reconnection and oscillation models, in which an MHD oscillation modulates the reconnection, resulting in periodic particle acceleration[31]. Due to the generally fast dynamics caused by reconnection and MHD oscillations, it is an ongoing challenge in observational astrophysics to temporally and spatially resolve such mechanisms.

Here we report on high-cadence (0.25 s) interferometric radio imaging of a pulsating radio source with a period of 2.3 s, observed by the Nançay Radioheliograph (NRH)[32], the Nançay Decametric Array (NDA)[33]; and the Orfées spectrograph (Observation Radio Fréquences pour l'Étude des Eruptions Solaires; a radio spectrometer operating at 100–1000 MHz). These radio observations are combined with EUV imaging from NASA's Solar Dynamics Observatory (SDO)[34]; and a model for the coronal magnetic field to show the locations of the pulsing radio emission among the magnetic topologies of the active region. The observations show evidence for an MHD sausage mode oscillation on a small magnetic loop located near a magnetic null point in the solar atmosphere. The MHD oscillation is associated with periodic electron acceleration at the null point. These electrons are then repeatedly injected into adjacent coronal loops, where they modulate a loss-cone instability, ultimately resulting in pulsing plasma emission. Overall, this shows that a complex interaction of MHD oscillations, electron acceleration and loss-cone instability modulation can ultimately produce pulsing radio emission in the solar atmosphere.

## Results

**Imaging of pulsating radio source in quiet active region.** On 18 April 2014, a flux rope eruption and M7.3 class flare commenced at 12:35 Universal Time (UT) from active region AR12036 (labelled flare active region (flare AR) in Fig. 1a), observed using the Atmospheric Imaging Assembly (AIA)[35]; on-board SDO.

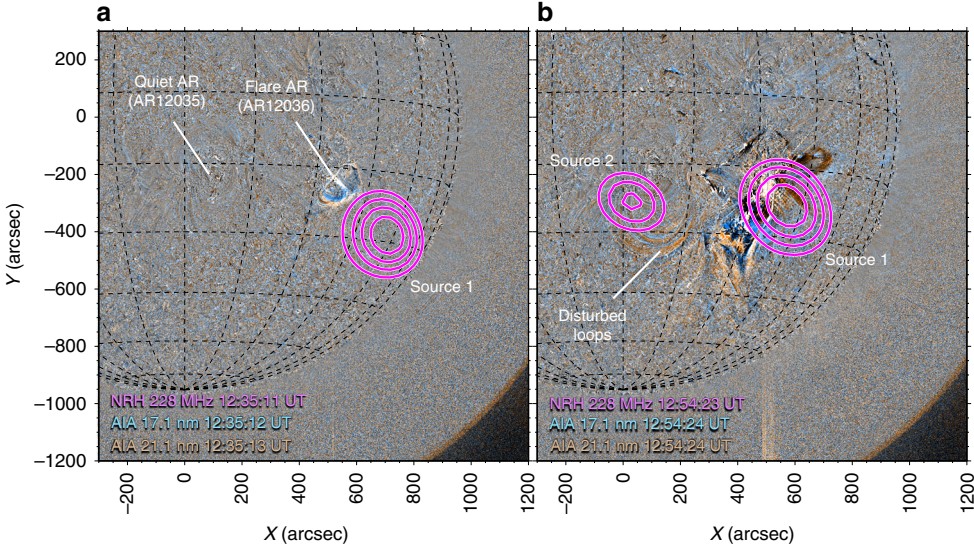

**Fig. 1** EUV and radio imaging of the flare and quiet active regions. **a** Extreme ultraviolet running ratio images from the SDO AIA 21.1 and 17.1 nm passbands overlaid with NRH 228 MHz contours. The flare AR is responsible for the majority of radio activity during the event, such as Source 1. **b** In a quiet AR approximately 400 Mm (600 arcseconds) east of the flare site, a second radio Source 2 appears. This quiet AR shows signs of disturbed coronal loops (as indicated) at the time of Source 2. Source 2 shows periodic variations in flux density for up to 2.5 min, shown in Fig. 2 below

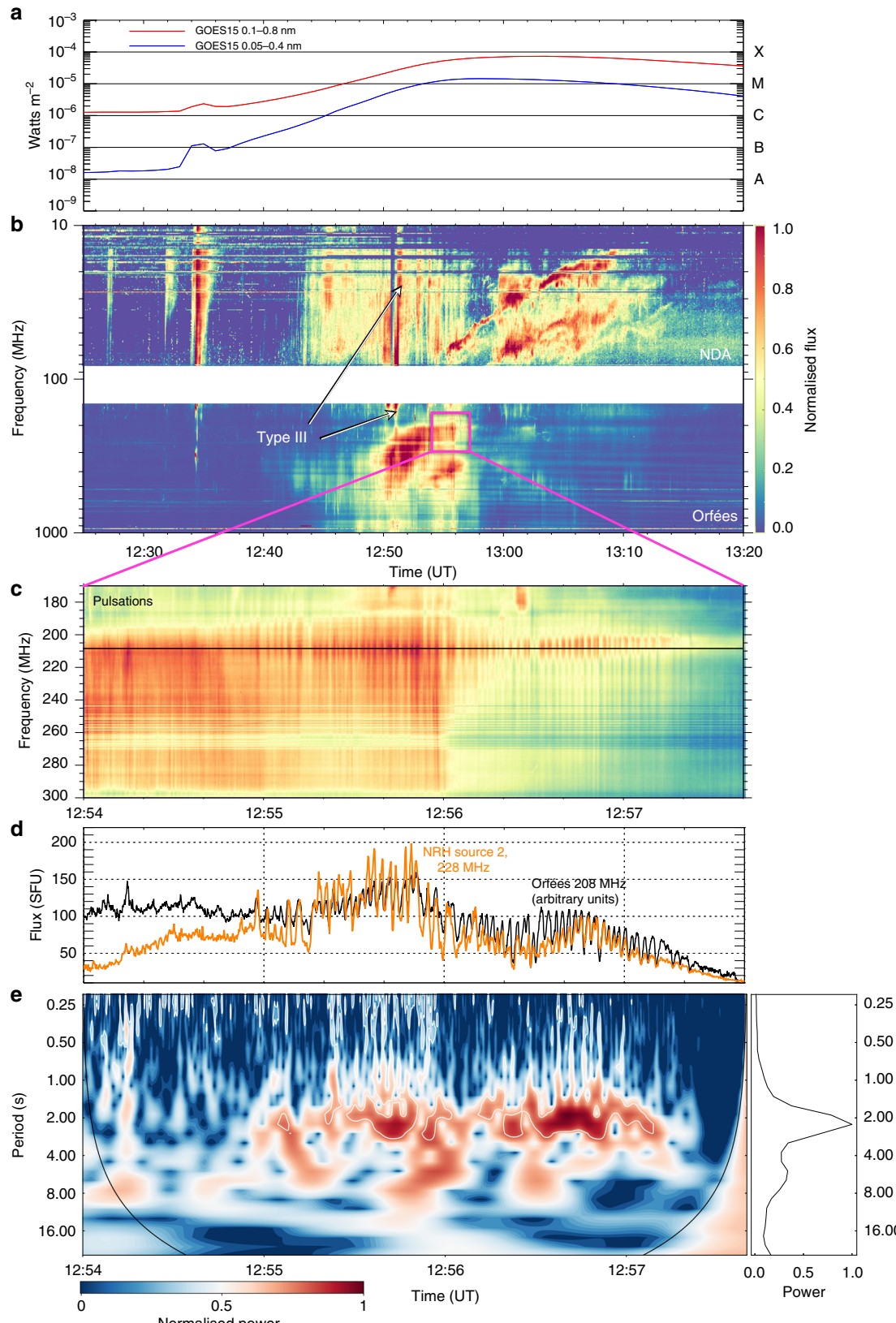

This eruption was associated with a variety of radio sources concentrated at the flare AR (see Source 1, Fig. 1a), the details of which can be found in[36]. Active region AR12035 (labelled quiet AR in Fig. 1a) was situated approximately 400 Mm east of the flare AR. Although the quiet AR appeared relatively stable with little sign of activity, a radio source was observed at this location, labelled Source 2 in Fig. 1b. At the time of Source 2, the quiet AR shows only signs of disturbed coronal loops (indicated in Fig. 1b), likely perturbed by the passing of an EUV wave eastwards at approximately 310 km s$^{-1}$, shown in Supplementary Fig. 1.

**Fig. 2** X-ray and radio activity associated with the event. **a** GOES soft X-ray light-curves, showing an M7.3 flare. **b** NDA and Ofrées dynamic spectra (time resolution of 1.0 and 0.1 s, respectively, with normalised flux indicated by the colour bar on the right) showing all radio bursts associated with the event (see[38] for an overview of this radio activity). For this article, we concentrate on the radio activity in the small purple box, a zoom of which is shown in panel **c** (normalised flux on the right colour bar). The pulsations are clearly seen from approximately 12:54:50 UT to approximately 12:57:30 UT extending over the range of approximately 190–300 MHz, and peaking at approximately 208 MHz. **d** A light-curve extracted from Orfées at 208 MHz (black) with the flux of Source 2 in NRH at 228 MHz in orange for comparison. The Source 2 pulsation and the Orfées pulsation show good agreement through time. **e** The wavelet spectrogram of the Orfées flux, displaying power (indicated by the bottom colour bar) that consistently remains at approximately 2.3 s

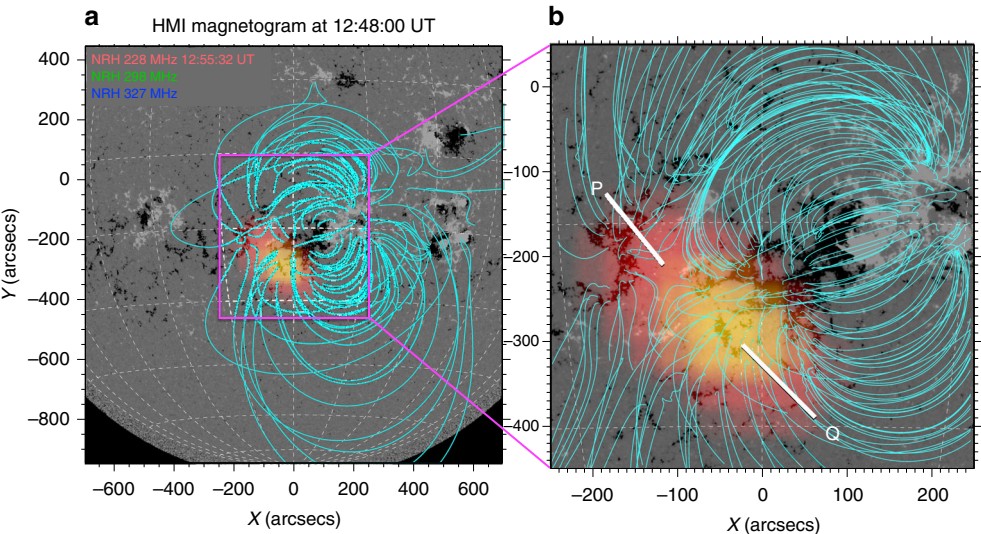

**Fig. 3** Pulsation radio source location in the active region magnetic field. **a** SDO HMI magnetogram with a PFSS extrapolation of the quiet AR magnetic field shown in blue. The pulsating radio source is overlaid as a three-colour image of NRH 228 MHz (red), 298 MHz (green) and 327 MHz (blue) at 12:55:32 UT. **b** A zoom of the active region with a NLFFF magnetic extrapolation in blue. At least part of the radio source is located at point P, with some emission extending into the closed field region of the magnetic loop footpoints, labelled Q. Note the region above the higher negative field strength in the magnetogram (black regions) shows greater intensity in the higher frequencies (yellow colouration), indicating that this emission is likely related to the magnetic field strength

Figure 2 shows the spectral radio activity associated with the radio images, displaying a variety of complex radio bursts between 10 and 1000 MHz in Orfées and the NDA; the majority of these radio bursts were associated with the flare AR[36]. Among this radio activity, from approximately 12:54 to 12:58 UT a set of radio pulsations are observed to occur at approximately 190–300 MHz, Fig. 2c. Figure 2d shows these pulsations are correlated in time with the flux of Source 2, indicating the quiet AR was the origin of the pulsations (radio activity in the quiet AR initially begins with a slow rise in radio flux from approximately 12:45 UT and a type III burst at 12:51 UT labelled in Fig. 2b, see Supplementary Note 1; here we concentrate primarily on the time of the pulsations). Using a Morlet mother wavelet[37] on the Orfées light-curve at 208 MHz, the pulsations are shown to have a steady 2.3-s period, see Fig. 2e. The power is narrowband showing a strong periodic component, unlike other solar flaring pulsations, which often show a quasiperiodicity, e.g., in X-ray observations[5,6]. This analysis shows that although the quiet AR displayed a relatively small amount of activity, there was the formation of a pulsating radio source with a steady periodicity of 2.3 s. In the following, we show where this radio source occurs in the magnetic topology of the quiet AR.

**Pulsating source location in active region magnetic field**. To show the location of the pulsating radio source in the quiet AR, we plot an NRH three-colour image on top of the Heliospheric and Magnetic Imager (HMI) magnetogram and use both a potential field source surface (PFSS) and non-linear force free

field (NLFFF) extrapolation of the active region magnetic field (see Methods), shown in Fig. 3a, b. The radio source is located close to the footpoints of the coronal loops in the west of the active region. Based on the NLFFF in Fig. 3b, we separate the region into point P (the location of a possible magnetic null point), and point Q, the area around the adjacent coronal loop footpoints. Although the 228 MHz emission (red areas) is spread from P to Q, the higher frequencies of 298, 327 MHz (yellow areas) mainly concentrate around Q. Also, the regions where the source transitions to yellow (higher frequencies) is concentrated above the negative magnetic field intensity, shown by the black areas in the underlying magnetogram. This suggests that the source emission mechanism may have a dependency on magnetic field strength.

Supplementary Movie 1 shows the behaviour of the pulsating radio source over time. The radio emission initially builds up at the coronal loops footpoints around Q, indicating an initial injection of energetic electrons into the region (also observed as increases in EUV, microwave and hard X-ray (HXR) activity from approximately 12:45 UT, see Supplementary Notes 1 and 2). When the pulsations begin in the dynamic spectrum, the radio source shows a repeated brightening at point P to the north-east. Supplementary Fig. 4 and Note 3 show it is mainly point P that pulses at 228 MHz, while the source at point Q shows some pulsing but a steadier variation in intensity from 228 to 327 MHz. From the movie, the radio source also appears to show a back-forth motion between P and Q, however, we show in Supplementary Note 4 that the motion is merely an apparent one, and is due to alternating brightness of the sources located at P and Q.

By observing a small sample of the pulses in detail, we are able to discern the relationship of the emission sources between P and Q. Figure 4 shows one pulse event in NRH images, along with a zoom of the dynamic spectrum and light-curves at 208, 298 and 327 MHz from Orfées (and the closest frequencies in NRH), which show a sawtooth-like pattern of slow rise and more rapid decay in intensity. Figure 4a, b shows that during the pulse rise and peak the 228 MHz emission increases at point P. During the pulse decay, the emission diminishes at P and mainly concentrates at Q (Fig. 4c). Throughout the process, the 298 and 327 MHz sources remain close to Q. The entire process lasts for approximately 2.3 s and repeats for each pulsation, see Supplementary Movie 2.

During the pulsation peak at P, we observe reverse drift bursts in the dynamic spectrum. These features drift at approximately 220 MHz s$^{-1}$ and become narrower with increasing frequency, similar in shape to the reverse drift bursts during flares[38], or the herringbone activity during type II radio bursts[39]. Such bursts are attributed to plasma emission generated by downward propagating electron beams in the corona, hence these features may indicate that the pulsations are at least partly due to electron beam acceleration at point P (the metric radio pulsation envelope is also well correlated with HXR and microwave flux, providing further evidence of association with energetic electrons, see Supplementary Fig. 3). Unfortunately, the separation of the radio sources at P and Q is close to the resolution limit of NRH at 228 MHz (approximately 100″), so we are unable to discern radio source motion due to beam propagation.

Although the reverse drifters and pulsing radio source at P suggest electron acceleration at this point, the steady state of the

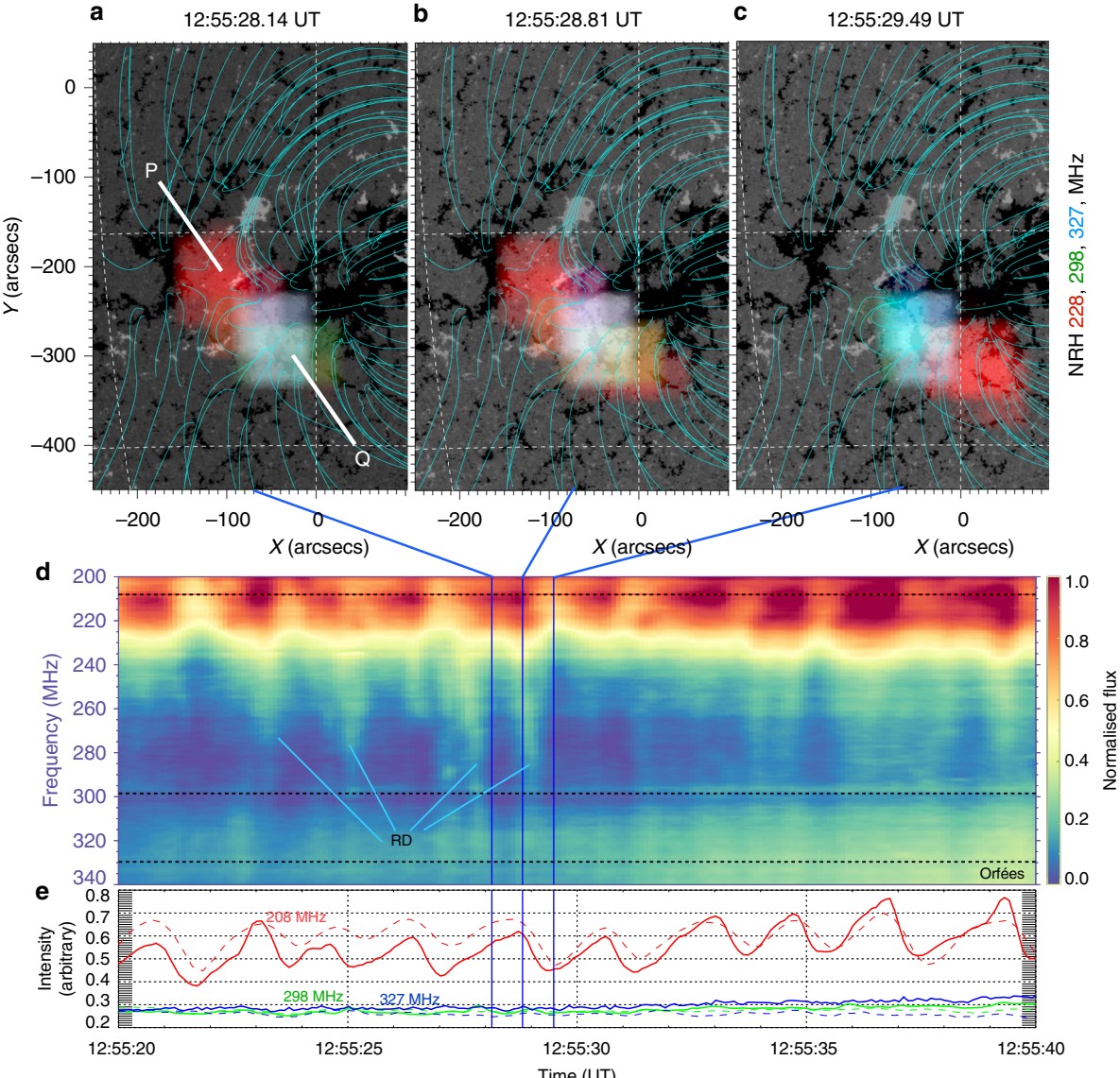

**Fig. 4** Radio source behaviour of one pulsation. **a–c** NRH sources at 228 MHz (red), 298 MHz (green) and 327 MHz (blue) overlaid on a HMI magnetogram, with NLFFF extrapolation shown in blue. **d** The dynamic spectrum shows a zoom of 10 pulsations, most of which have reverse drift bursts or signatures of electron beams. **e** The intensity profiles are from Orfées (solid lines) and the closest frequencies in NRH (dashed lines), showing a sawtooth pattern at 208 MHz (Orfées) and 228 MHz (NRH). During the rise phase of the sawtooth, 228 MHz emission increases at P. Starting at the emission peak, and at the time of the reverse drifters (labelled RD) formation in the dynamic spectrum, the 228 MHz source begins to elongate towards Q. During pulse decay the 228 MHz source diminishes at P. During the process, the higher frequencies remain steady in intensity and position around Q. The process lasts for approximately 2 s and then repeats with a new pulsation, see Supplementary Movie 2

298–327 MHz source at Q would suggest a concentration of energetic electrons at coronal loops footpoints (a magnetic trap) around Q. We also note that electron acceleration at P takes place` at a magnetic field configuration that represents a magnetic null point, i.e., there are four separate magnetic domains that meet at this point and curve away from each other. This is observed both in the NLFFF extrapolation and also in the shape of the field as determined from EUV emission at 17.1 nm in Supplementary Fig. 6. In the following, we examine the magnetic field extrapolations and the spectral characteristics of the radio emission to determine both the cause of the periodic activity and how this leads to modulated radio emission.

**MHD sausage mode oscillations**. One of the primary candidates for the production of periodic pulsations is through modulation of the radio emission from MHD oscillations. In such a scenario, the plasma properties such as density, pressure and magnetic fie`ld strength modulates any emission mechanism that depend on such properties. Here we test if the pulsing emission at P can be be due to an MHD oscillation.

For the scenario of MHD oscillation modulation of the radio emission, the observed period of 2.3-s coincides with the predicted timescales of fast sausage mode MHD oscillations in coronal conditions[40,41]. To test whether MHD sausage modes could be responsible for the radio pulsations we search the pulsating radio source origin for a candidate coronal loop that could sustain such an oscillation. Unfortunately loop dynamics at a period of approximately 2.3 s cannot be imaged at the cadence of AIA (12 s). However, we find a candidate loop for the sausage mode oscillations at the null point centre at the position where the pulsing radio sources originate, see Fig. 5a, b. Such a magnetic loop would sustain a sausage mode oscillation with a period given by $P_{\mathrm{saus}} \lesssim \frac{a}{C_{\mathrm{f}}}$[41,42], where $a$ is the loop radius and $C_{\mathrm{f}} = \sqrt{v_{\mathrm{A}}^2 + c_{\mathrm{s}}^2}$ (fast magnetosonic speed) where $v_{\mathrm{A}}$ is the Alfvén speed and $c_{\mathrm{s}}$ is the thermal sound speed given by

$$v_{\mathrm{A}} = \frac{B}{\sqrt{4\pi m_{\mathrm{p}} n_{\mathrm{p}}}} \tag{1}$$

$$c_{\mathrm{s}} = \sqrt{\frac{\gamma k_{\mathrm{B}} T}{m_{\mathrm{p}}}} \tag{2}$$

Here $B$ it the magnetic field strength, $m_{\mathrm{p}}$ and $n_{\mathrm{p}}$ are the mass and density of protons, respectively, $k_{\mathrm{B}}$ is the Boltzmann constant, $T$ is the loop temperature and $\gamma$ is the adiabatic index (taken to be 5/3 in this case). First, to estimate density we perform a differential emission measure (DEM) inversion of the six AIA filters (9.4, 13.1, 33.5, 19.3, 21.1, 17.1 nm) by implementing the method outlined in[43]. We perform the inversion on a small region around the null point, producing DEM maps of the region, which we then sum across temperature to obtain a total emission measure (EM) map as shown in Fig. 5b. To obtain a measure of electron density from this EM map, we use $n_{\mathrm{e}} = \sqrt{EM/l}$, where $l$ is the line-of-sight depth through the loop. We make the assumption that the loop is axially symmetric and its sky-plane width $w$ is equal to its depth $l$. Figure 5b indicates the lines across which we estimate the width $w$ (distance between the inner and outer green points along each radial trace through the loop). We calculate $n_{\mathrm{e}}$ for each direction through the loop such that we obtain density as a function of distance along the loop, shown in Fig. 5d (uncertainty on the EM value is taken to be approximately 10%[43] and assuming a loop width uncertainty of 10% the resulting density uncertainty is approximately 7%). As a measure that the density values are physically reasonable,

we have fitted a model for a hydrostatic equilibrium along both legs (indicated by the purple and blue lines in Fig. 5d). The procedure shows that the loop has a radius of between 2.5 and 5.5 Mm, length of approximately 35 Mm, density of between $n_{\mathrm{e}} = 1.7$–$2.7 \times 10^9$ cm$^{-3}$, and temperatures of between 1 and 1.6 MK, giving a thermal sound speed of 111–145 km s$^{-1}$.

Next, we use the NLFFF extrapolation to estimate the loop magnetic field strength. Figure 5c shows the three-dimensional (3D) volume of extrapolated magnetic field lines at the null point. The loop of interest is too small to be extrapolated specifically in the NLFFF; to determine its 3D shape we make the assumption that the distance from the origin (point o in Fig. 5b) to the loop central axis (blue points Fig. 5b) is equal to its height, giving the 3D extrapolated loop position marked by blue points in Fig. 5c. For a measure of the magnetic field in the loop environment, we take an average value in a $20 \times 20 \times 20$ Mm cube around the 3D loop volume (blue box in Fig. 5c), resulting in average field of 49 G. Using this field strength and the density found from the DEM analysis, we obtain an Alfvén speed in this region of 2070–2605 km s$^{-1}$. Combining this Alfvén speed with the sound speed and loop width calculated above, we find that the loop would sustain a sausage mode oscillation of $P \leq 1.3 - 1.6$ s, which matches well with the observed period of $2.3 \pm 1.0$ s (we show in Methods that given the quoted fractional uncertainties for density and loop width above, the fractional uncertainty on this period is equal to the magnetic field fractional uncertainty). Finally, to estimate the position of the radio source with relation to this small loop, we assume the 228 MHz is plasma emission with a frequency of $f_{\mathrm{p}} \sim 8980\sqrt{n_{\mathrm{e}}}$ (where $n_{\mathrm{e}}$ is the electron density in cm$^{-3}$), then use a hydrostatic density model for the local environment using a base density of $n_{\mathrm{e}} = 10^9$ cm$^{-3}$ and temperature of 1 MK (values of same order of magnitude from the EM analysis of the small loop). We find an altitude of 13 Mm for fundamental emission and 55 Mm for harmonic emission (as indicated in Fig. 5c), which are within the vicinity of the small loop. Also, given the drift rate of the reverse drifters (220 MHz s$^{-1}$) and density model derived here, the exciter speed of these bursts is approximately 0.2 c, which provides further evidence that they are from electron beams propagating to higher density regions.

This analysis indicates the presence of a loop with physical properties that would produce an MHD sausage oscillation of period $P \leq 1.3 - 1.6$ s in close proximity to a radio source that pulses at a period of $2.3 \pm 1.0$ s, indicating that a sausage mode oscillation modulation of the radio emission is at least a feasible one. There are, however, alternative mechanisms that can produce the steady periodicities we observe in this event. The other primary candidate is known as a Lotka-Volterra or predator–prey system, in which electron and wave energy distributions exchange energy in cyclic fashion. In Supplementary Fig. 7 and Note 5, we show that the characteristics of the pulsations here do not match a Lotka-Volterra system, making the MHD oscillations hypothesis the more likely candidate in producing the pulsations.

Overall, this shows that an MHD oscillation in the vicinity of the null point was related to the radio emission pulsation. The simultaneous observation of reverse drift bursts (as well as the correlated HXR and microwave flux) at the same time as MHD oscillations suggests the sausage mode has a relationship with electron acceleration. In the following, we examine the emission characteristics in detail to establish which emission mechanisms are at play and how the MHD oscillation and electron acceleration brings about the pulsations.

**Emission mechanism and loss-cone instability modulation**. Although the reverse drift bursts indicate the presence of plasma

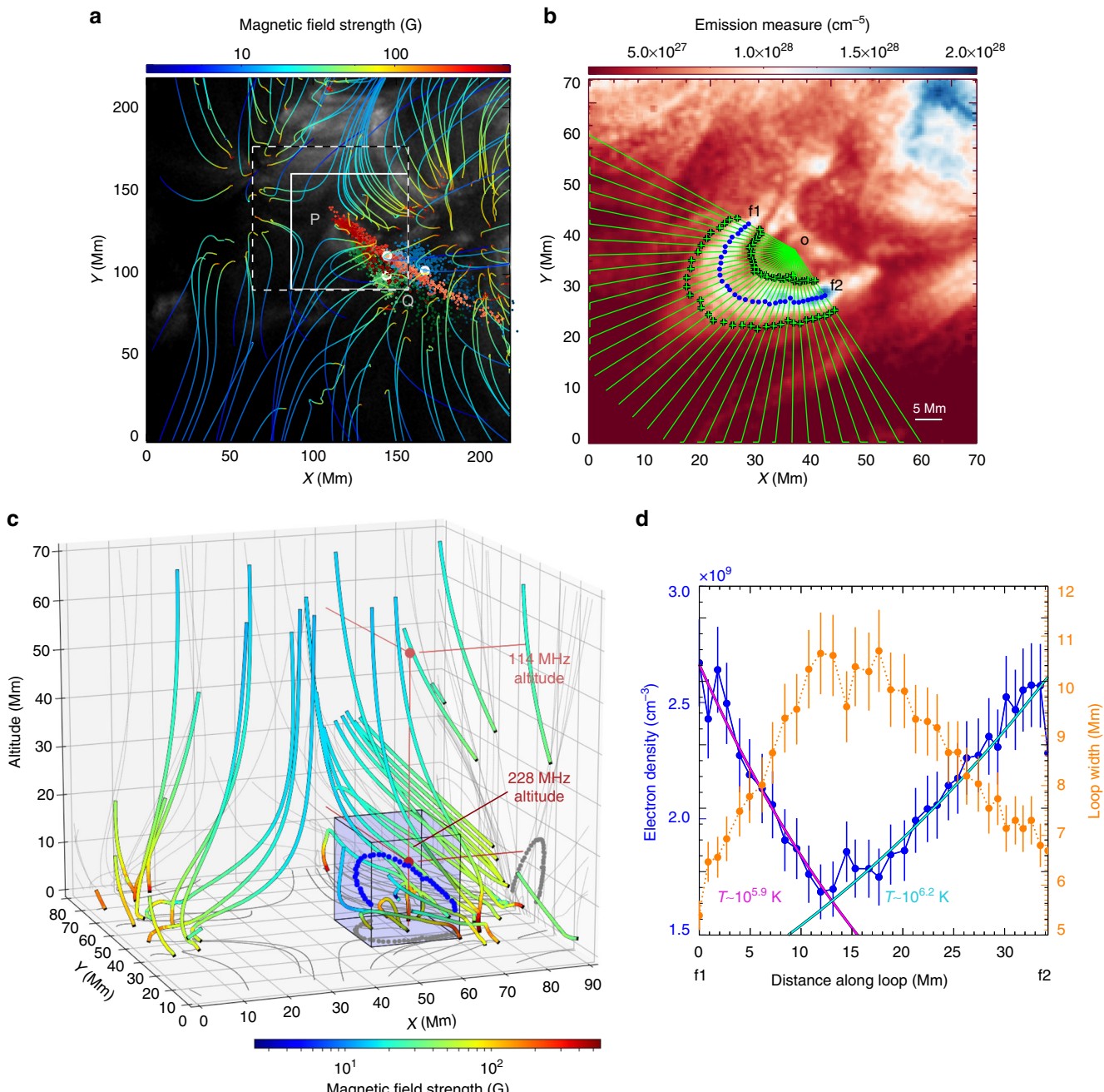

**Fig. 5** Magnetic field and density environment of the pulsation origin. **a** NLFFF extrapolation on AIA 33.5 nm, with the 228, 298 and 327 MHz source maxima throughout the event in red, green and blue points, respectively. **b** Emission measure map of a zoom of the null point centre (white box in panel a), where we identify a small loop. The width of the loop (distance between inner and outer green points for each radial trace starting at point o) is calculated from f1 to f2. **c** 3D environment of the NLFFF extrapolation (white dashed box in panel **a**), with the 3D position of the small loop indicated by the blue points and the plasma emission heights of 114 and 228 MHz indicated by red points. **d** Density as a function of distance along the loop (blue circles, with uncertainty of 7%) from f1 to f2. The purple and blue lines are models of a hydrostatic equilibrium with temperature of $10^{5.9}$ K and $10^{6.2}$ K, respectively. The orange data points indicate the loop width as a function of distance along the loop (with uncertainty of 10%). Using the NLFFF magnetic field strength and the density estimate from the EM map, the loop in panel **b** is capable of sustaining sausage mode oscillation period of $P \lesssim 1.3$–$1.6$ s

emission due to electron beams, the pulsations also have a separate spectral characteristic that indicate a secondary mechanism is at play. Figure 6a shows the flux density spectrum of a single pulse in detail, showing the spectrum at pulse peak at 12:56:32 UT and pulse trough (1 s later). The pulse has a large flux density of approximately 900 SFU (non background-subtracted), with a narrow bandwidth of approximately 30 MHz. The spectral indices on the positive and negative slopes are

particularly extreme, with values of $\alpha_+ = 21$ and $\alpha_- = -13$, respectively. Observation of such large spectral slopes are rare, but previous studies have provided rigorous theoretical explanations of these extreme spectral characteristics[23,44]; they conclude the mechanism that bests accounts for such a spectral shape is plasma emission from electrostatic waves produced by a loss-cone instability, known as the Z-mode of the electron cyclotron maser (ECM) instability[45–51]. In Supplementary Note 6, we show

that the coronal conditions and a variety of characteristics of the radio emission (emission location, spectral slopes, Stokes-V polarisation (see Supplementary Figure 5), and ratio of plasma to gyrofrequency $\omega_e/\Omega_e$) are all supportive of the ECM Z-mode, hence indicating the presence of a loss-cone at the pulsation site.

Figure 6b also shows that there is a modulation of slopes $\alpha_+$ and $\alpha_-$, in phase with the modulation in intensity. Since these spectral indices likely depend on the loss-cone instability that generates the radio emission, modulation of the indices may indicate modulation of the instability. In previous studies, it was proposed that pulsations in solar flares may be a result of emission firstly being generated via a loss-cone distribution in coronal loops, followed by this emission being modulated by the repeated injection of energetic electrons into the loops[52–54]. The modulation would occur by the electron beam quenching the loss-cone instability and causing emission sudden reductions[16,23] (see Supplementary Fig. 8 and Note 7 for a discussion of the

sudden reductions in this event). Such a mechanism is possible here and explains the observations of reverse drift bursts (electron beam injection) immediately followed by sudden reduction (loss-cone quenching).

The above analysis has shown evidence for simultaneously occurring MHD sausage mode oscillations, electron acceleration, and a loss-cone instability modulation. This suggests a complex interaction of these mechanisms in bringing about the radio pulsations, which we outline in the final Discussion section.

**EUV wave kinematics.** While the above show evidence for a variety of mechanism involved in the pulsation activity in the event, it is unclear as to what triggers the activity in the first place. The region shows little sign of flaring, but does show a perturbation of the active region loops toward the south-east in Fig. 1. One possibility for the cause of this perturbation is the impact of the EUV wave on the quiet AR. Figure 7 shows a kinematic

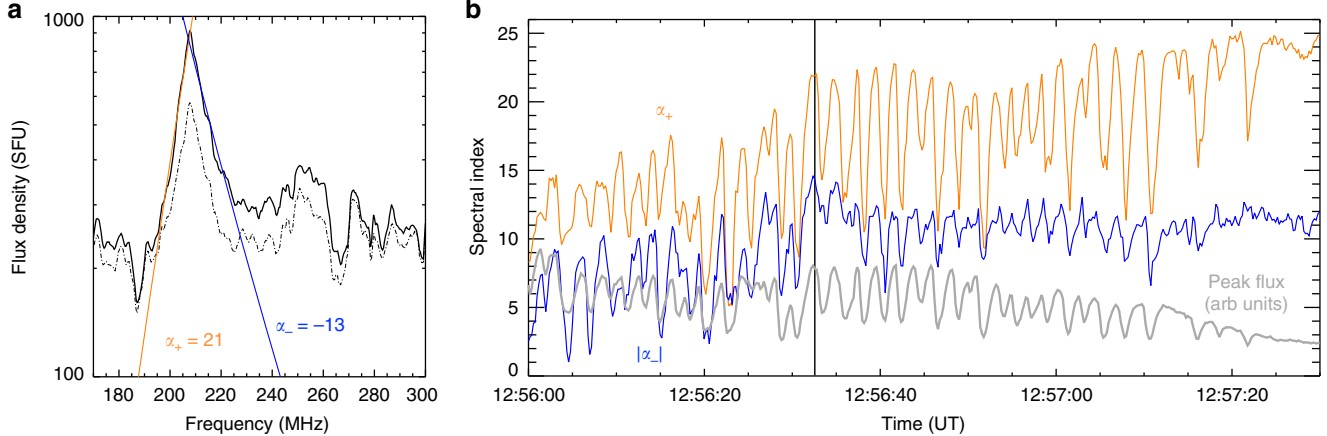

**Fig. 6** Pulsation flux density spectrum properties throughout time. **a** Flux density spectrum of a single pulse taken from the Orfées dynamic spectrum, showing pulse maximum at 12:56:30 UT in solid and pulse trough (approximately 1 s later) in dot-dash. Fits to both the positive (orange) and negative (blue) slopes of the pulse at max intensity indicate spectral indices of $\alpha_+ = 21$ and $\alpha_- = -13$. **b** The spectral indices over time for a subset of the pulsations from 12:56:00 to 12:57:30 UT, showing that the pulsing in intensity is also accompanied by a pulsing in the spectral indices

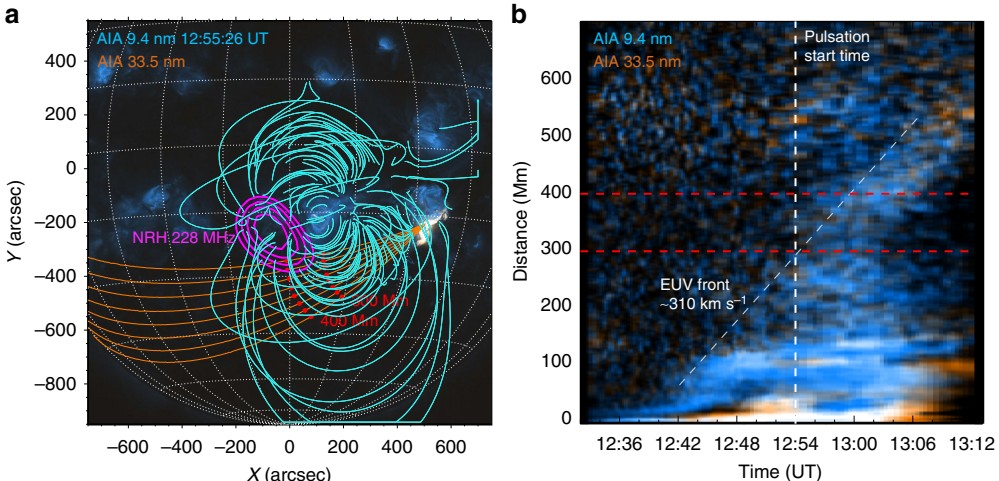

**Fig. 7** Kinematics of the EUV perturbation. **a** Two colour image of AIA 9.4 nm (blue) and 33.5 nm (orange). The blue lines are the PFSS extrapolation of the quiet AR, with the contours of the pulsating 228 MHz radio source in purple. The orange lines are great circles originating at the flaring active region and spreading across the disk towards the quiet active region and beyond. The red circles indicate 300 and 400 Mm along each great circle. Intensity traces are taken along these great circles and summed to produce a distance-time map. **b** Distance-time map from the great circles marked in panel **a**. The distances of 300 and 400 Mm are also marked on this map by the red dashed lines. The pulsation start time is marked by the vertical dashed line. A perturbation can be seen propagating along the great circles to larger distances. This front has a speed of 310 km s$^{-1}$ and reaches the distance of the radio source (approximately 300 Mm) around the start time of the pulsations at approximately 12:54 UT

analysis of the propagation of the EUV perturbation across the disk in the direction of the quiet AR. The perturbation is faint in the cooler channels of 17.1, 19.3 and 21.1 nm, but in the hotter 9.4 and 33.5 nm channels a running ratio distance-time (*d-t*) map (where each time is divided by the time 60 s before). The map shows the EUV disturbance begins propagating eastward just after approximately 12:42 UT, at which time it reaches the east side of the quiet AR (at approximately 100 Mm in the *d-t* map). This time coincides with the initial flux increase in metric radio, HXR, microwave and EUV activity, see Supplementary Notes 1 and 2. The EUV perturbation then reaches the distance of the pulsating source (approximately 200 Mm) at the time the pulsations begin (approximately 12:54 UT). The large-scale EUV perturbation may have played a role in increasing the activity in the quiet AR, eventually resulting in the pulsations. MHD oscillations generated in loops impacted by a passing large-scale disturbance has been suggested in a previous pulsation event[4]. Ref. [55] has also shown that the passage of a high-amplitude disturbance past a 3D magnetic null can cause collapse of the null and the non-linear growth of oscillatory reconnection, specifically mentioning that an EUV wave could be such a disturbance. The impact on the quiet AR of the erupting structure from the flare remains a possibility in explaining why the pulsating radio source occurred in the quiet AR. As for the duration of the pulsation event (approximately 2.5 min), it is unclear why the pulsations show no immediate signs of damping (as might be expected for a sausage mode oscillation). This could be due to the prolonged disturbance of the region supplied by the passage of the EUV wave over the course of several min, as in[4] for example. However, the means by which the EUV wave triggers and sustains oscillatory behaviour at the null point is beyond the scope of this study.

## Discussion

In this analysis, we have shown evidence for repeated electron acceleration close to a magnetic null resulting in emission in the form of reverse drift bursts. At the time of the repeated electron acceleration, we find evidence for an MHD sausage mode oscillation on a small loop near the null point. The MHD oscillation and electron acceleration occur simultaneously with a modulation of ECM Z-mode emission generated from a loss-cone instability. The variety of physical mechanisms at play suggests they have a complex interaction in bringing about the radio pulsations.

A physical scenario that explains this complex interaction is outlined in Fig. 8. First, there is an initial establishment of a loss-cone instability at magnetic footpoints between the P and Q sites (Fig. 8a). This would require an initial acceleration of electrons in the quiet AR and is evidenced by the initial radio source from 12:45 UT seen in Supplementary Movie 1, as well as the simultaneous increase in microwave, HXR, and EUV emission from 12:45 UT onwards (as shown in Supplementary Figs. 2 and 3). Next, there is a sausage mode oscillation excited on a small loop near P (see Fig. 8b), evidenced by the presence of a magnetic loop with physical properties that could sustain a sausage mode period on the order of seconds (same as the radio pulsations). The sausage mode results in a pulsed electron acceleration, observed as the reverse drift bursts. As we have evidence for electron acceleration close to field lines that resemble a null point, magnetic reconnection is a possible mechanism for the acceleration. MHD oscillation triggering reconnection at a null point has been suggested previously[5,31,56], and can happen either via fast waves leaking from the oscillating loop or by density variations near the null point.

The accelerated electrons are injected onto loops around P and Q, see Fig. 8b. They propagate downwards, temporarily filling the loss-cone on these loops, quenching the instability and leading to sudden reductions in the radio emission spectrum. The loss-cone emission is evidenced as spectral characteristics indicating ECM Z-mode and the emission location at magnetic footpoints between the P and Q sites, see Supplementary Notes 6.

Pulsations due to loss-cone quenching from repeated electron injection into a magnetic trap has been reported in previous

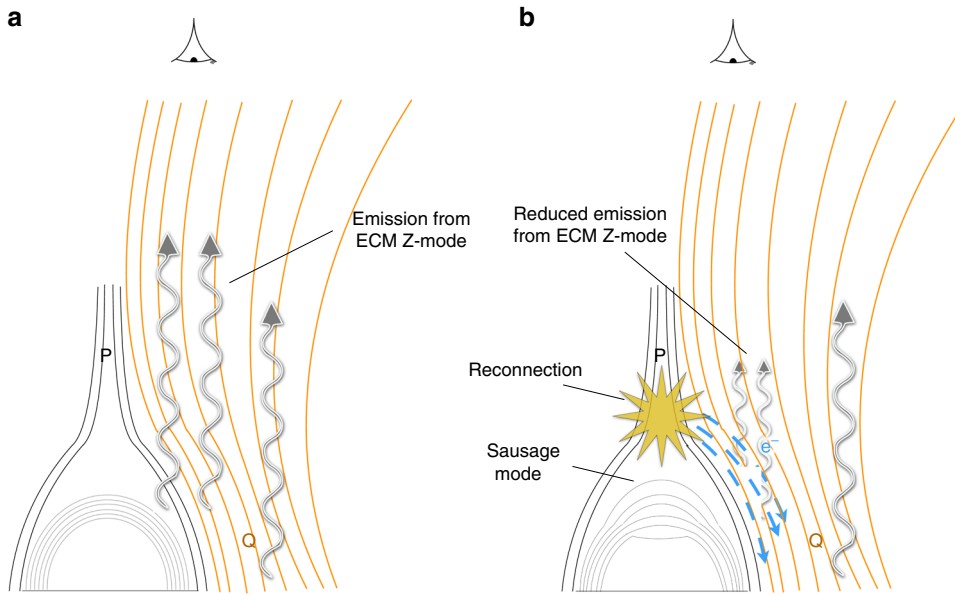

**Fig. 8** Schematic of physical scenario in operation to produce the pulsations. **a** Electrons accelerated in the region initially form a loss-cone distribution at magnetic footpoints located around P and Q, resulting in loss-cone plasma emission (ECM Z-mode). **b** Establishment of an MHD sausage mode oscillation on the small loop results in acceleration of electrons at P (possibly via reconnection) due to modulation of the plasma conditions around the null point. Propagation of these electrons away from the null point (blue arrows) lead to reverse drift bursts from plasma emission. When the electrons reach the surrounding footpoints at P and Q, they quench the loss-cone instability, resulting in reduction of loss-cone emission. Some of the emission around Q remains unaffected. This would result in a radio emission modulation at the period of the sausage mode

studies[23,54,57,58], with[16] in particular showing similar pulsation and spectral characteristics to this event. In this case, the process is observed using 2D radio interferometry combined with magnetic field extrapolation. We also observe evidence for both the beam acceleration/injection (reverse drifters) immediately followed by the loss-cone quenching (reduction in the spectrum intensity and indices during the decay phase). Note that while the pulsing emission occurs between P and Q at 228 MHz, the higher frequency emission of 298–327 MHz at Q shows little sign of pulsation. This may imply emission around Q comes from a broader range in heights, with sources lower in the atmosphere (at 298–327 MHz) unaffected by the electron injection, e.g., in Fig. 8b electrons do not pass through some of the emission around Q, so it remains unaffected.

Overall, the pulsations are a repetition of sausage mode oscillation, electron beam acceleration, followed by loss-cone quenching. The entire process would result in a modulated radio emission with a period prescribed to the sausage mode, which originally triggers the electron beam. Because the MHD oscillation is not a direct modulator of the radio emission, the pulse profile does not need to be symmetric or sinusoidal and may explain its sawtooth characteristic in Fig. 4. For example, the pulse sawtooth rise phase would include the build up of loss-cone emission (and at certain times direct observation of the reverse drift beam), while the sawtooth decay phase would be due to loss-cone quenching. If the loss-cone emission growth timescale were to be longer than the emission decay timescale (from instability quenching), the pulse profile would be asymmetric. In Supplementary Note 7, we show that such asymmetric growth and decay timescales can be accounted for using theoretical timescales for particle injection, plasma wave damping, and plasma wave re-growth.

For alternative scenarios leading to periodic radio emission, we cannot specifically rule out the action of a Lotka-Volterra system, but given the pulsation characteristics this mechanism has less supportive evidence. Periodic magnetic reconnection itself has also been suggested for radio pulsations, however, such systems are expected to be either quasiperiodic and variable[27], or highly damped oscillations[55], neither of which match the observations in this event. The future development of radio instrumentation should aim at an ability to observe this process at high-time resolution with full imaging spectroscopy. This will provide further advances in the understanding of oscillations and particle instabilities on the Sun, in astrophysical systems and in Earth-based laboratories.

## Methods

**Magnetic field extrapolations and three-colour radio maps**. The active region large-scale magnetic field was modelled firstly from a PFSS extrapolation at 12:04 UT (just before flare occurrence) using the standard methods outlined in[59,60], and overplotted on the HMI[61]; magnetogram. For a more detailed modelling of the field around the radio sources, we used a NLFFF extrapolation. This was performed on HMI data at 12:00 UT using the weighted optimisation method, initially proposed by[62] and implemented by[63]. NLFFF extrapolations are generally considered to more accurately represent small-scale coronal magnetic field structures than the PFSS, which are a first-order approximation to the coronal magnetic topology[64]. HMI magnetogram data (zonal, meridional, and radial components) were used as inputs to the 3D magnetic field extrapolation, with the data preprocessed using the procedure outlined by[65]. Although a range of estimates for the loop properties is possible given the range of parameters of width, temperature, density and magnetic field, a true estimate of uncertainty is not possible due to the lack of uncertainty analysis from the NLFFF magnetic magnetic field values[65]. That said, given the expressions for the sausage mode period $P$, Alfvén speed $v_A$, and sound speed $c_s$, propagation of uncertainty gives

$$\frac{\delta P}{P} = \sqrt{\left(\frac{\delta a}{a}\right)^2 + \left(\frac{v_A \delta v_A}{v_A^2 + c_s^2}\right)^2 + \left(\frac{c_s \delta_{c_s}}{v_A^2 + c_s^2}\right)^2} \tag{3}$$

$$\frac{\delta v}{v} = \sqrt{\left(\frac{\delta B}{B}\right)^2 + \left(\frac{\delta n_p}{2 n_p}\right)^2} \tag{4}$$

$$\frac{\delta c_s}{c_s} = \sqrt{\left(\frac{\delta T}{2T}\right)^2} \tag{5}$$

where $B$, $n_p$ and $T$ are magnetic field strength, proton number density and temperature, respectively. With uncertainties $\delta n_p / n_p \sim 0.07$, $\delta a / a \sim 0.1$ and $\delta T / T \sim 0.2$, we find that $\delta P / P \sim \delta B / B$, so the period uncertainty is not highly sensitive to the magnetic field intensity uncertainty, e.g., 20% magnetic field uncertainty yields the same period uncertainty.

The radio sources were plotted on top of the magnetogram as a three-colour image; the radio maps are first individually normalised to their own maximum and then plotted together as 228 MHz (R; red), 298 MHz (G; green) and 327 MHz (B; blue) in an RGB image; any white regions indicate a co-spatial intensity peak from all three NRH frequencies, while a primary colour (or combination of primaries) indicates one (or two) frequencies reaching their respective peak intensity at these locations. We emphasise that because the maps are normalised before combinations, white or primary colour combinations do not indicate areas of equal intensity, but areas where all three or two frequencies reach a peak co-spatially.

## Data availability

The radio datasets analysed during the current study are available in the Nançay Radioheliograph archive at http://secchirh.obspm.fr/nrh_data.php. Any AIA or HMI datasets are available at the SDO archive https://sdo.gsfc.nasa.gov/data/. Any custom datasets that support the findings of this study are available from the authors upon reasonable request.

## Code availability

The magnetic field extrapolation software packages are available through IDL SolarSoft at https://sohowww.nascom.nasa.gov/solarsoft/. Any custom code used to produce figures in the article can be found at https://github.com/eoincarley/pulsationshttps.

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

## Acknowledgements

E.P.C. was supported by ELEVATE: Irish Research Council International Career Development Fellowship—co-funded by Marie Curie Actions, and is currently supported by the European Commission Horizon 2020 INFRADEV-1-2017 LOFAR4SW project, no. 777442. L.A.H. is supported is supported by Enterprise Partnership Scheme studentship from the Irish Research Council (IRC) between Trinity College Dublin and Adnet System Inc., S.A.M. is supported by the Irish Research Council Postdoctoral Fellowship Programme and the Air Force Office of Scientific Research award number FA9550-17-1-039. D.E.M. acknowledges external funding from Met office, United Kingdom and the Finish Centre of Excellence in Research for Sustainable Space (Academy of Finland grant no. 1312390). We are grateful to the SDO and GOES teams for open access to their data. The NRH is funded by the French Ministry of Education and the Région Centre. Orfées is part of the FEDOME project, partly funded by the French Ministry of Defense. The authors acknowledge the Nançay Radio Observatory/ Unité Scientifique de Nançay of the Observatoire de Paris (USR 704-CNRS, supported by Université d'Orléans, OSUC, and Région Centre in France) for providing access to NDA observations accessible online at http://www.obsnancay.fr. N.V. would like to acknowledge the support of CNES and of the French Program on Solar-Terrestrial Physics of INSU (PNST). We would like to thank Hamish Reid for some useful discussion during the writing of this paper. We would like to thank the referees for their useful comments and suggestions.

## Author contributions

E.P.C. wrote the article and performed the data analysis of radio imaging, radio dynamic spectra, EUV imaging, PFSS extrapolations, DEM analysis and part of the NLFFF analysis. L.A.H. performed the wavelet analysis. S.A.M. performed the NLFFF extrapolations. D.E.M., W.S., N.V. and P.T.G. provided advice on scientific interpretation and analysis.

## Additional information

**Competing interests:** The authors declare no competing interests.

