## [Peer Review File · Nature Communications]

Review of manuscript "Modulation of a loss-cone instability due to a MHD sausage mode oscillation in the solar corona" by Carley et al., submitted to Nature Communications

The author present some very interesting observations of a solar radio pulsation event that exhibits a very fast (2 s) period and interpret it in terms of a loss-cone instability that is coupled to an MHD sausage mode oscillation. The authors perform a wavelet analysis which convincingly shows a periodicity during most time of the flare. Although similar events have been reported earlier, what is unique in this study is that the authors have high-resolution EUV images from AIA/SDO and the tool of magnetic modeling at hand, which helps them to identify an X-point geometry that plays a keyrole in the interpretation. The data quality is very good and the Figures are of highest quality and very informative. Having said all these positive aspects, I think that the interpretation could be substantially improved, and therefore I suggest that the authors do some more in-depth work before the manuscript is considered for publication.

Essentially there are at least two interpretations: (A) MHD sausage mode, and (B) nonlinear limit cycles. The authors should discuss these two models in more detail, otherwise we do not learn anything in this paper. The first model could be possible if: (A1) the periodicity is very regular (as expected for a standing wave in an eigen-mode), (A2) if the modulated radio flux amplitude is sinusoidal with symmetric pulse shapes (as it is expected for a MHD sausage mode in the linear regime), and (A3) if the sausage mode period ($P_{\text{sausage}} > 2.62 a / v_A$) is consistent with the observables a and v_a . The loop diameters a could be measured from the AIA images, and the Alfvén velocity v_A can be obtained from the magnetic field (using the NLFFF method at the oscillating location) and the electron densities n_e (using a differential emission measure analysis, and possibly a hydrostatic density model). The possible range of the observables (a, n_e) should be determined and a mean value and standard deviation of the predicted sausage mode period P_{sausage} could be derived, including measurement uncertainties.

The second model (B) could also be modeled in more detail, which is completely absent in the present version. The authors mention a Lotka-Volterra equation system with a limit cycle for the energy exchange between plasma particle and wave distribution. In such a model, the wave growth (of the observed radio flux) has an e-folding growth time t_g and a decay time t_d for emptying the loss-cone from particles, while the limit cycle period has the mathematical solution of $P = \text{square root} (t_g * t_d)$, which is just the geometric mean of the growth and decay time. It is also straightforward to calculate the modulated wave flux profile $W(t)$ from the Lotka-Volterra equation and to show whether it can produce the non-symmetric pulse shapes, which look like sawtooth profiles as shown in Fig. 4. One can also use the NLFFF magnetic field model to calculate the mirror ratio of the loss-cone in the X-point region of the magnetic field. Together with some simplified transport model for emptying the loss cone, one can derive the decay time t_d . If the authors interpret the radio flux in terms of electron cyclotron maser emission, they could use the theoretically calculated growth times for the

various X, O, and W modes.

Once the authors have quantified the two models by using all the constraints they have from the (excellent) observations, they should be able to present a decisive discussion which model fits the data better. There are a number of possible criteria which the authors have not used. For instance, sausage (or kink-mode) mode oscillations usually exhibit very periodic but strongly damped time profiles, while limit cycles can be periodic over much longer time intervals, with occasional intermittent glitches (see Froment et al. 2015 for instance). In addition, are the saw-tooth pulses more consistent with sausage modes or with limit cycles ?

Some details:

The text could be shortened in a number of places. For instance the first half of the abstract says nothing new, the abstract could just start at: "Here we report on ..."

Introduction, "... but this has never before been achieved ..."
This is an overstatement

Introduction: "... modern challenge ..."
Not only in modern times !

Results: The acronyms or instruments Orfees and NDA are never explained

Many other acronyms are not defined or at the wrong places:
PFSS, NLFFF,

The term "thorns" should probably be "sawtooth pattern"

The term "pulse crash" probably means "pulse decay phase"

Discussion: "This is not a region where we would expect a loss-cone instability". Really ?

A magnetic field of $B > 39$ G is cited. Is this the average field strength along a loop? The corona is 3-dimensional, a single 0-dimensional value is useless, unless it is explained over what volume this value has been averaged.

Reviewer #2 (Remarks to the Author):

This paper reports an intriguing observation of radio oscillations that occurred during a solar flare. The radio source is attributed to emission produced by a loss-cone instability modulated by an MHD sausage mode oscillation in a compact loop near a magnetic null point. It is a very interesting observation but I am not convinced that the authors have demonstrated the proposed physical scenario.

The scenario is complex and involves several steps. A large flare and flux rope eruption occurred in AR12036. The radio oscillations were excited ~400 Mm to the east of the flare in AR12035.

- The authors argue that the EUV wave produced by the flare excited MHD oscillations in a compact, hot magnetic loop located near a magnetic null point.
 - o It is not clear from Supplementary Fig. 1 that the EUV wave (shown at 13:03 UT) in fact excited MHD oscillations at the time in question (1254-1258). The non-pulsating radio emission begins in this source at ~1245. The role of the EUV wave as an exciter needs to be shown more carefully.
 - o While the authors identify an EUV loop near the location of the oscillating radio source that could be the oscillator, they cannot show that the loop is in fact oscillating because the SDO cadence is too slow.
- The authors argue that loop oscillations periodically modulate magnetic reconnection and electron energization near the null point. This is asserted by the authors but cannot be shown.
 - o If the loop is implicated in reconnection, shouldn't it show some signs of being "activated" in radio or EUV emissions because fast electrons have access to the oscillating loop? Here, too, the timing and sequence of events is important.
- The authors claim that electrons are injected into an adjacent coronal loop that acts as a magnetic trap, presumably as part of the reconnection process – but it is left unstated.
 - o Into which magnetic loops might the electrons be injected based on the location of the null and the NLFF extrapolation? What is the mirror ratio of these loops? What is the height of the pulsating radio source?
- Next, a loss cone electron distribution results in the trap that is unstable to the production of Langmuir waves.
 - o Perhaps, but why would the loss cone be set up near the oscillating loop instead of the conjugate magnetic footpoint? Is the adjacent loop/trap itself compact? The authors raise the idea that there are electron beam signatures in the spectrum (not obvious to this reader). What is the role of electron beams in their scenario? By claiming spectral signatures of electron beams, the beams are themselves producing type III-like signatures? And loss-cone driven type IV? Do they

attribute loss-cone quenching to the purported electron beams? If so, why are the claimed beam-like signatures in phase with the pulse peaks?

- The authors claim that the Langmuir wave are then converted to radio waves that escape to the observer.
 - o The authors argue that the emission is plasma radiation. The Hewitt et al (1985) reference they provide is a formal study of the mechanism in an unmagnetized plasma (no polarization). The degree of polarization from harmonic plasma emission is expected to be weak (Melrose 1980), not the 70% degree of circular polarization reported here. The evidence is by no means “conclusive” that the suggested mechanism is operative. If the emission is harmonic plasma radiation the density in the source is $1.5 \times 10^8 / \text{cm}^3$. Is that commensurate with the plasma environment at the apparent height of the radio source?
 - o The authors go on to discuss the spectral index of the radio pulses. The relevance of references to work about loss-cone driven electron cyclotron maser emission was not clear to me.

In summary, each of the steps described in the overall scenario is plausible at some level but by no means inevitable. The observational evidence supporting each step is more suggestive than conclusive and, in some cases, completely absent. The overall scenario is speculative. Given the fact that remote pulsations occurred as part of a large flare, erupting flux rope, and fast halo CME, aren't other means of energizing and distributing electrons over a wide volume, and exciting MHD or other periodic phenomena, equally plausible?

Reviewer #3 (Remarks to the Author):

The manuscript presents a detailed study of quasi-periodically pulsating radio emission generated in the solar corona, supplemented by the study of the magnetic field and plasma geometry in the source region. The phenomenon of quasi-periodic pulsations is now at the forefront of the ongoing research effort in solar and stellar physics, which makes the results presented in the paper especially timely. The main novel element of the paper is the use of the combination of spatially-resolving observations in radio and EUV, and the state-of-the-art theory. It allowed to authors to trace the whole physical chain of the induction of the pulsations, and use it for a convincing interpretation of the observed phenomenon. In my opinion, the paper is a very solid and timely contribution to the research field of solar radio physics and plasma physics, and should be published as soon as possible.

I would suggest some minor revision of the text.

Abstract: "In rare cases, ..." - I would suggest to modify this statement, as quasi-periodic pulsations of the radio emission intensity are a rather frequent phenomenon.

p. 2: "... known as a Lotka-Volterra system" - better, "modelled by ..."

" Solar radio pulsations usually display periodicities of ~0.3-4 seconds" - those pulsations have much longer periods too. I'd suggest to rephrase this statement.

p. 17, the estimation of the sausage mode oscillation period. I would remove the factor of 2.62, as recent works (including the cited in this estimations) demonstrated that this factor appears only in the case of a loop with a step-function perpendicular profile of the fast speed, while for smoother profiles this factor is different. Hence, a more correct estimation would be P less than a/C_F , where C_F is the fast speed. But, this modification does not affect the result and conclusions.

Reviewer #1 (Remarks to the Author):

Comment 1:

Review of manuscript "Modulation of a loss-cone instability due to a MHD sausage mode oscillation in the solar corona" by Carley et al., submitted to Nature Communications. The author present some very interesting observations of a solar radio pulsation event that exhibits a very fast (2 s) period and interpret it in terms of a loss-cone instability that is coupled to an MHD sausage mode oscillation. The authors perform a wavelet analysis which convincingly shows a periodicity during most time of the flare. Although similar events have been reported earlier, what is unique in this study is that the authors have high-resolution EUV images from AIA/SDO and the tool of magnetic modeling at hand, which helps them to identify an X-point geometry that plays a keyrole in the interpretation. The data quality is very good and the Figures are of highest quality and very informative. Having said all these positive aspects, I think that the interpretation could be substantially improved, and therefore I suggest that the authors do some more in-depth work before the manuscript is considered for publication.

Response:

We would like to thank the referee for their useful comments and suggestions on our article. We have significantly improved the interpretation with a more comprehensive analysis of the physical mechanisms at play in the event. The second half of the paper is rewritten, and we have added four new Figures, namely Figure 6 and 8 in the main article text and Supplementary Figures 4 and 5. Details responses to the referee's comments are given below. We believe the new article more accurately demonstrates the proposed physical scenario and addresses all of the referee's concerns. All changes to the article text are in blue.

Comment 2:

Essentially there are at least two interpretations: (A) MHD sausage mode, and (B) nonlinear limit cycles. The authors should discuss these two models in more detail, otherwise we do not learn anything in this paper. The first model could be possible if: (A1) the periodicity is very regular (as expected for a standing wave in an eigen-mode), (A2) if the modulated radio flux amplitude is sinusoidal with symmetric pulse shapes (as it is expected for a MHD sausage mode in the linear regime), and (A3) if the sausage mode period ($P_{sausage} > 2.62 a / v_A$) is consistent with the observables a and v_a . The loop diameters a could be measured from the AIA images, and the Alfvén velocity v_A can be obtained from the magnetic field (using the NLFFF method at the oscillating location) and the electron densities n_e (using a differential emission measure analysis, and possibly a hydrostatic density model). The possible range of the observables (a , n_e) should be determined and a mean value and standard deviation of the predicted sausage mode period $P_{sausage}$ could be derived, including measurement uncertainties.

Response:

We have performed a much more comprehensive analysis of the MHD sausage mode oscillation concept. The new analysis is outlined in the new 'MHD sausage mode oscillations' section and now includes a new Figure 6 in the main article. We have followed the referee's recommendations and have performed a much more accurate analysis of the loop geometry, as well as a differential emission measure (DEM) analysis to obtain density. We have also exploited the 3D information provided by the NLFFF extrapolation to estimate a more accurate estimate of the magnetic field in the region of the pulsations (at the small loop). This comprehensive analysis now shows the loop physical characteristics (width, density, temperature and magnetic field) can produce a sausage mode oscillation of period range of $P < 1.3 - 1.6$ seconds. This matches quite well the observed period 2.3 ± 1.0 seconds of the pulsations and means the sausage mode oscillation idea is feasible.

While a range of estimates for the loop properties is possible given the range of parameters of width, temperature, density and magnetic field, a true estimate of uncertainty is not possible due to the lack of uncertainty analysis from the NLFFF magnetic field values (Wiegmann et al.

2012). That said, given the relationship of P to the magnetic field (via Alfvén speed), their fractional uncertainties are approximately equal e.g, $\Delta P = \Delta B$, so a 20% uncertainty in B will lead to ~20% uncertainty in P. Hence P is not highly sensitive to the B uncertainty.

Comment 3:

*The second model (B) could also be modeled in more detail, which is completely absent in the present version. The authors mention a Lotka-Volterra equation system with a limit cycle for the energy exchange between plasma particle and wave distribution. In such a model, the wave growth (of the observed radio flux) has an e-folding growth time t_g and a decay time t_d for emptying the loss-cone from particles, while the limit cycle period has the mathematical solution of $P = \sqrt{t_g * t_d}$, which is just the geometric mean of the growth and decay time. It is also straightforward to calculate the modulated wave flux profile $W(t)$ from the Lotka-Volterra equation and to show whether it can produce the non-symmetric pulse shapes, which look like sawtooth profiles as shown in Fig. 4.*

Response:

While the pulse profiles may be fit using solutions of the differential equations of a Lotka-Volterra system, we find that the physical characteristics of the pulse do not match with theoretical predictions of such a system. Several authors have shown that the relationship between the pulse period t and pulse flux F for an LV system should be $F \sim t^2$ (Aschwanden and Benz 1988, Fleishman et al. 1994). Using a peak finding algorithm, we find that no such relationship exists for the pulsation data, with F and t being uncorrelated, see Supplementary Figure 4. Furthermore, the LV system is likely to result in a more quasiperiodic pulsation and larger variability in the flux. The pulsations presented in this event are more steady and slowly varying. This does not specifically rule out such a mechanism being at play, but it makes it less likely than the MHD oscillations scenario. The above arguments are now stated in a new Supplementary Material section entitled ‘Likelihood of a Lotka-Volterra system’.

Comment 4:

One can also use the NLFFF magnetic field model to calculate the mirror ratio of the loss-cone in the X-point region of the magnetic field. Together with some simplified transport model for emptying the loss cone, one can derive the decay time t_d . If the authors interpret the radio flux in terms of electron cyclotron maser emission, they could use the theoretically calculated growth times for the various X, O, and W modes.

Response:

Using some of the magnetic field estimates from the NLFFF we have included a more complete discussion on wave growth and particle decay time, and the consequences for loss-cone mirror ratios and distances travelled by the particles to reach the loss-cone, as outlined in the new Supplementary Material section ‘Viability of the sudden reduction mechanism’. This discussion, along with Figure 6c, now gives a better idea of the physical conditions involved in creating the loss cone.

As for the mode of radiation, several authors have shown the circumstances under which we may observe X, O, and Z-modes in response to a loss-cone instability. The X and O modes are generally only employed in situations where $\omega_e/\Omega_e < 1$ (the limiting value varies slightly depending on the energetic electron distribution involved), where Ω_e is electron cyclotron frequency and ω_e is electron plasma frequency. Such conditions are only likely to be met low in the corona, where the magnetic field is strong and emissions reach millimetric wavelengths. The Z-mode however has high growth rate in regions of $\omega_e/\Omega_e > 1$. Based on this ratio and the more accurate density and magnetic field analysis, we provide further theoretical basis for the interpretation of the emission as being from the Z-mode. This is outlined in the ‘Emission mechanism and loss-cone instability modulation’ section of the main article and also in the new ‘Dominant ECM modes’ section of Supplementary Material.

The referee mentions the possibility of the W-mode (assumed the referee means Whistler mode from the ECM). Indeed this is a possibility but there is not much evidence for the existence of Whistlers in the explanation of narrow-band pulsating radio sources. Their growth rate is high for ratios $\omega_e/\Omega_e > 1$. In this instance, the magnetic field would need to be 81 G in order to produce a whistler of frequency Ω_e . From the NLFFF extrapolation, this is likely to occur deeper in the corona below <10 Mm where this ratio is unlikely to be fulfilled.

Comment 5:

Once the authors have quantified the two models by using all the constraints they have from the (excellent) observations, they should be able to present a decisive discussion which model fits the data better. There are a number of possible criteria which the authors have not used. For instance, sausage (or kink-mode) mode oscillations usually exhibit very periodic but strongly damped time profiles, while limit cycles can be periodic over much longer time intervals, with occasional intermittent glitches (see Froment et al. 2015 for instance). In addition, are the saw-tooth pulses more consistent with sausage modes or with limit cycles ?

Response:

The two new sections 'MHD sausage mode oscillations' and 'Likelihood of a Lotka-Volterra system' now outline the feasibility of the two physical mechanisms and which of the two is more likely. We find the MHD sausage oscillation mechanism to be more likely in this instance. As we discuss in the conclusions, if the sausage mode oscillation were to directly modulate the emission (via modulation of the plasma properties), we would expect more symmetric pulse profiles. However, because the mechanism is indirect (sausage-mode, electron beam formation, loss-cone quenching), this does not necessitate a symmetric pulse profile. We have also remade Figure 4 such that the 'reverse drift bursts' (previously called "thorns") are now seen more clearly. We have also analysed these RD bursts, showing they have the same characteristics to previous observations of reverse drifters that occur from electron beam propagation (Aschwanden et al. 1995). This provides further evidence of the idea that electron acceleration was involved in the pulsations, meaning the modulation of radio emission was indirect. This points are outlined in the discussion section.

As for the duration of the pulsation event (2.5 minutes), it is unclear why these pulsations last so long (showed no strong damping). This could be due to the prolonged disturbance of the region supplied by the passage of the EUV wave, as in McLean et al. (1971) for example. Passage of an EUV wave in bringing about oscillatory dynamics of a null point has been recently highlighted by Thurgood et al. (2017). However the exact involvement of the EUV wave in bringing about such activity in this event remains unclear. These points are highlighted in the 'EUV wave kinematics section'.

Some details:

Comment 6:

The text could be shortened in a number of places. For instance the first half of the abstract says nothing new, the abstract could just start at: "Here we report on ..."

Response:

We would prefer to keep the first half of the abstract. It gives a brief outline of the background phenomena in question, which is helpful for the general reader. As for the text in general, we have made it as brief as possible by moving some sections to the Supplementary material. The discussion and conclusion section has been shortened.

Comment 7:

Introduction, "... but this has never before been achieved ..."

This is an overstatement

Response:

Agreed, statement has been removed.

Comment 8:

Introduction: "... modern challenge ..."

Not only in modern times !

Response:

Agreed, changed to "ongoing challenge"

Comment 9:

Results: The acronyms or instruments Orfees and NDA are never explained

Response:

Acronyms now defined and the instrument paper is provided for NDA. Orfées does not have an instrument paper, but we briefly mention what the instrument is.

Comment 10:

Many other acronyms are not defined or at the wrong places:

PFSS, NLFFF,

Response:

Acronyms now defined.

Comment 11:

The term "thorns" should probably be "sawtooth pattern"

Response:

The sawtooth pattern is applicable to a single frequency. However, the pattern we indicate is formed over multiple frequencies. We have replaced the word 'thorns' with the more general 'reverse drift bursts', which is a more generic term to describe signatures which quickly drift to higher frequencies. We have also analysed these bursts showing they have the characteristics of previously observed reverse drift bursts that are signatures of electron beam propagation (Aschwanden et al. 1995).

Comment 12:

The term "pulse crash" probably means "pulse decay phase":

Response:

Agreed, we have replaced 'crash' with 'decay phase'

Comment 13:

Discussion: "This is not a region where we would expect a loss-cone instability". Really ?

Response:

The discussion has been re-written and this statement removed.

Comment 14:

A magnetic field of $B > 39$ G is cited. Is this the average field strength along a loop? The corona is 3-dimensional, a single 0-dimensional value is useless, unless it is explained over what volume this value has been averaged.

Response:

With the more accurate determination of magnetic field in the NLFFF extrapolation (Figure 6) we have now quoted the volume in which we average the magnetic field. The volume is now shown as a blue box in Figure 6c of the article.

Reviewer #2 (Remarks to the Author):

Comment 1:

This paper reports an intriguing observation of radio oscillations that occurred during a solar flare. The radio source is attributed to emission produced by a loss-cone instability modulated by an MHD sausage mode oscillation in a compact loop near a magnetic null point. It is a very interesting observation but I am not convinced that the authors have demonstrated the proposed physical scenario.

Response:

We would like to thank the referee for their useful comments and suggestions on our article. We have significantly improved the interpretation with a more comprehensive analysis of the physical scenario at play in the event. The second half of the paper is rewritten, and we have added four new Figures, namely Figure 6 and 8 in the main article text and Supplementary Figures 4 and 5. Details responses to the referee's comments are given below. We believe the new article more accurately demonstrates the proposed physical scenario and addresses all of the referee's concerns. All changes to the article text are in blue.

Comment 2:

The scenario is complex and involves several steps. A large flare and flux rope eruption occurred in AR12036. The radio oscillations were excited ~400 Mm to the east of the flare in AR12035.

- *The authors argue that the EUV wave produced by the flare excited MHD oscillations in a compact, hot magnetic loop located near a magnetic null point.*

- o *It is not clear from Supplementary Fig. 1 that the EUV wave (shown at 13:03 UT) in fact excited MHD oscillations at the time in question (1254-1258). The non-pulsating radio emission begins in this source at ~1245. The role of the EUV wave as an exciter needs to be shown more carefully.*

Response:

Agreed. We have provided a more careful kinematic analysis that shows the expansion of an EUV disturbance along great circles from the erupting active region, see Figure 8. The disturbance reaches the location of the pulsating radio source at the time the pulsations begin. This provides an indication that the the EUV disturbance was involved in the perturbation of the region that caused the pulsations.

Comment 3:

- o *While the authors identify an EUV loop near the location of the oscillating radio source that could be the oscillator, they cannot show that the loop is in fact oscillating because the SDO cadence is too slow.*

Response:

Agreed. It is generally not possible to image such fast pulsations in EUV imaging. Despite this, we have performed a much more comprehensive analysis of the MHD sausage mode oscillation concept. The new analysis is outlined in the 'MHD sausage mode oscillations' section and now includes a new Figure 6 in the main article. We have performed a much more accurate analysis of the loop geometry, as well as a DEM analysis to obtain density. We have also exploited the 3D information provided by the NLFFF extrapolation to estimate a more accurate estimate of the magnetic field in the region of the pulsations (at the small loop). This comprehensive analysis now shows us that the physical location of the pulsating radio source in 3D dimensions was next to a loop whose physical characteristics (width, density, temperature and magnetic field) can produce a sausage mode oscillation of period range of $P < 1.3 - 1.6$ seconds. This matches quite well the observed period 2.3 ± 1.0 seconds of the pulsations and means the sausage mode oscillation idea is feasible.

We have also tested a second primary method by which pulsations could be generated, namely the Lotka-Volterra system. Several authors have shown that the relationship between the pulse period t and pulse flux F for an LV system should be $F \sim t^2$. Using a peak finding algorithm, we find that no such relationship exists for the pulsation data, with F and t being uncorrelated. Furthermore, the LV system is likely to result in a more quasiperiodic pulsation and larger variability in the flux. The pulsations presented in this event are more steady and slowly varying. This does not specifically rule out such a mechanism being at play, but it makes it less likely than the MHD oscillations scenario.

Comment 4:

- *The authors argue that loop oscillations periodically modulate magnetic reconnection and electron energization near the null point. This is asserted by the authors but cannot be shown.*

Response:

Agreed. It cannot specifically be shown that electron acceleration was via magnetic reconnection. We have reserved mention of such a mechanism to the discussion section, where we state it remains a possible scenario due to electron acceleration being located at a magnetic null. That said, lack of evidence of magnetic reconnection does not change the general conclusion of our paper that there is loss-cone modulation via an MHD oscillation.

Comment 5:

- o *If the loop is implicated in reconnection, shouldn't it show some signs of being "activated" in radio or EUV emissions because fast electrons have access to the oscillating loop? Here, too, the timing and sequence of events is important.*

Response:

While the loop showing some motion or brightening in the EUV images would be good evidence for its involvement, there does not necessarily need to be a signature of the loop activation. For example, electrons accelerated in the null-point region may not have access to the small loop. If no energetic particles enter the small loop it would not show significant signs of heating/flaring. Indeed the loop shows little sign of activation throughout the event, apart from one leg being slightly hotter than the other (see Figure 6d). The slightly hotter leg may be a signature of heating at its base due to chromospheric evaporation, but such a conclusion can be only tentative. Nonetheless, we believe the analysis we have provided shows the MHD oscillation modulation of emission is a feasible one.

Comment 6:

- *The authors claim that electrons are injected into an adjacent coronal loop that acts as a magnetic trap, presumably as part of the reconnection process – but it is left unstated.*

Response:

We have provided a more thorough analysis and discussion on this idea including an estimate of where the electrons are accelerated (Figure 6c). This now gives a clearer picture of the environment in which the acceleration took place. Specifically identifying the trap location is not possible due to the limited spatial resolution of the radio imaging. However, there are multiple locations within the magnetic environment of the pulsations where a magnetic trap and loss-cone could form e.g., on the adjacent magnetic footpoints (around Q) or at any of the other magnetic footpoints around the null point. This is discussed further in the Discussion & Conclusion section.

Comment 7:

- o *Into which magnetic loops might the electrons be injected based on the location of the null and the NLFF extrapolation? What is the mirror ratio of these loops? What is the height of the pulsating radio source?*

Response:

As stated in the previous comments, it is generally not possible to pin-point the exact injection or reflection point of the electrons due to the limited spatial resolution of the radio imaging. Despite this, the analysis performed in Figure 6 gives a much clearer indication of the environment of the pulsating radio source in three dimensions. Based on the analysis of the radiation being from Langmuir waves due to Z-mode ECM emission (or from stream instability plasma emission in the

case of the reverse-drift bursts), we have also provided an estimate of the height of the radio source as being 13 Mm for fundamental emission, or 55 Mm for harmonic (indicated in Figure 6c). This places it within the vicinity of the small loop. Based on the magnetic field estimates in this vicinity, particle injection and plasma wave growth times, we have outlined a discussion in the Supplementary Material section ‘Feasibility of the sudden reduction mechanism’ which discusses particle injection, plasma wave growth times, mirror ratios and estimated heights of the radiation.

Comment 8:

- *Next, a loss cone electron distribution results in the trap that is unstable to the production of Langmuir waves.*
 - o *Perhaps, but why would the loss cone be set up near the oscillating loop instead of the conjugate magnetic footpoint? Is the adjacent loop/trap itself compact?*

Response:

The idea in this instance is electron acceleration at the null point, with these electrons propagating into the adjacent footpoint regions, which could be adjacent footpoint at Q. This idea is very similar to the studies of Benz et al. (1976), Aschwanden et al. (1993, 1995). The reverse drift bursts are likely the signature of initial electron acceleration (via plasma emission) while the injection of these electrons into the trap is evidenced by sudden reduction of loss cone emission. It is not possible to estimate a size for this trap, as it is likely smaller than the radio source sizes. The most information we have is that the trap at Q is smaller than the radio source size of 200 arcseconds at this location. That said, absence of trap size estimate does not affect our calculations and conclusions. We outlined a discussion of particle injection into a trap in Discussion & Conclusions section.

Comment 9:

The authors raise the idea that there are electron beam signatures in the spectrum (not obvious to this reader). What is the role of electron beams in their scenario? By claiming spectral signatures of electron beams, the beams are themselves producing type III-like signatures? And loss-cone driven type IV? Do they attribute loss-cone quenching to the purported electron beams? If so, why are the claimed beam-like signatures in phase with the pulse peaks?

Response:

We have reprocessed the zoomed dynamic spectrum in Figure 4 so that the features are more clearly seen (now labelled simply as ‘reverse drift bursts’ or RD). We have also provided new analysis and references to show they have the same drift rate (220 MHz/s) and morphology as the reverse drift bursts studied previously, similar to Aschwanden et al. (1995). Their exciter speed is calculated to be 0.2 c using the hydrostatic density model derived in the ‘MHD sausage mode oscillations’ section. In light of this new evidence, it is now more clear that these features are from electron beams propagating into greater densities (down a magnetic loop) in the corona. Their emission mechanism is likely to be standard plasma emission generated from Langmuir waves driven unstable from an electron beam (as in the standard quasi-linear theory for type IIIs). Once these electron beams are injected into an adjacent trap, they result in sudden reduction of the loss-cone emission that is taking place in the trap. The reduction in emission is a combination of the time taken for electrons to be injected into the trap and the time taken for the decay of plasma waves via collisions — this idea is largely the same as the Benz et al. (1976) study. Therefore, it is possible for the signature of the beam and sudden reduction to be out of phase due to one following the other in quick succession. We have provided a more detailed explanation of this in ‘Emission mechanism and loss-cone instability modulation’, ‘Discussion & Conclusions’ and a calculation of the feasibility of the idea in the Supplementary Material ‘Viability of the sudden reduction mechanism’ section.

Comment 10:

- *The authors claim that the Langmuir wave are then converted to radio waves that escape to the observer.*
 - o *The authors argue that the emission is plasma radiation. The Hewitt et al (1985) reference they provide is a formal study of the mechanism in an unmagnetized plasma (no polarization). The degree of polarization from harmonic plasma emission is expected to be weak (Melrose 1980), not*

the 70% degree of circular polarization reported here. The evidence is by no means “conclusive” that the suggested mechanism is operative. If the emission is harmonic plasma radiation the density in the source is $1.5 \times 10^{18} / \text{cm}^3$. Is that commensurate with the plasma environment at the apparent height of the radio source?

Response:

While the Hewitt et al. (1985) references mentions small levels of polarisation, there are several other studies which mention it could be quite large e.g., Melrose et al. (1978) showing that second harmonic plasma radiation from Langmuir waves generated in a loss-cone distribution may be X polarised depending on the direction of Langmuir waves, viewing angle, and loss-cone angle. Using their Figure 2 and 3 and using $\omega_e / \Omega_e = 1.6$ and loss-cone angle of 30 degrees from our event we find polarisation can be X mode polarised from ~50-100%, depending on viewing angle. We have now quoted these references in ‘Emission mechanism and loss-cone instability modulation’.

In order to conclude the mechanism is operative we use several characteristics including the source location, spectral indices, flux, narrow bandwidth, polarisation and ratio of plasma to gyro frequency. The extreme spectral indices in particular are good evidence in favour of plasma emission from the ECM Z-mode mechanism.

As for the height of the source, we have used the plasma frequency and its harmonic to estimate density then a hydrostatic equilibrium to estimate height of such a density. Assumption of fundamental emission places the radio source height 13 Mm, while assumption of harmonic emission places it at 55 Mm, both of which place the radio source within the vicinity of the small loop (see Figure 6c).

Comment 11:

The authors go on to discuss the spectral index of the radio pulses. The relevance of references to work about loss-cone driven electron cyclotron maser emission was not clear to me.

Response:

In Benz et al. (1976a, 1976b) the authors discuss the phenomenon of some narrowband radio bursts having very large spectral slopes of >10 . They discuss the challenges in the theoretical explanation of such extreme spectral characteristics and specifically rule out synchrotron radiation. Synchrotron spectral slopes are usually an order of magnitude lower than this, even in cases of significant Razin suppression. The authors then discuss the most likely theoretical scenario to produce spectral indices of this magnitude, concluding that such a spectrum is from ‘plasma waves in transverse direction to the magnetic field...excited by trapped particles due to the presence of a loss cone’ i.e., the ECM Z-mode. The event in this study is another rare example of a radio burst having extreme spectral slopes and the evidence we provided corroborates and supports the previous interpretations of this emission being from electrostatic waves generated by a loss cone instability. We have reworded the text in ‘Emissions mechanism and loss-cone instability modulation’ so the reference to previous studies is now more clear.

Comment 12:

In summary, each of the steps described in the overall scenario is plausible at some level but by no means inevitable. The observational evidence supporting each step is more suggestive than conclusive and, in some cases, completely absent. The overall scenario is speculative. Given the fact that remote pulsations occurred as part of a large flare, erupting flux rope, and fast halo CME, aren't other means of energizing and distributing electrons over a wide volume, and exciting MHD or other periodic phenomena, equally plausible?

Response:

We have provide more complete and robust evidence on the interpretation of the physical scenario we have postulated, namely the operation of an MHD wave in the modulation of the radio emission. We have also included an assessment of another primary mechanism (the Lotka-Volterra system), showing that this mechanism is less likely. It is possible that the erupting CME may have been involved in triggering the pulsations (as is evidenced from the arrival of the EUV

disturbance), but it is unlikely that the the radio source belonged to the CME itself. Carley et al. (2016) Figure 10 has shown that the legs of the CME (where one might expect a loss-cone, for example) were rooted in the flaring active region and not connected to the pulsing source.

Reviewer #3 (Remarks to the Author):

The manuscript presents a detailed study of quasi-periodically pulsating radio emission generated in the solar corona, supplemented by the study of the magnetic field and plasma geometry in the source region. The phenomenon of quasi-periodic pulsations is now at the forefront of the ongoing research effort in solar and stellar physics, which makes the results presented in the paper especially timely. The main novel element of the paper is the use of the combination of spatially-resolving observations in radio and EUV, and the state-of-the-art theory. It allowed to authors to trace the whole physical chain of the induction of the pulsations, and use it for a convincing interpretation of the observed phenomenon. In my opinion, the paper is a very solid and timely contribution to the research field of solar radio physics and plasma physics, and should be published as soon as possible.

I would suggest some minor revision of the text.

Response:

We would like to thank the referee for their useful comments and suggestions on our article. We have significantly improved the interpretation with a more comprehensive analysis of the physical mechanisms at play in the event. The second half of the paper is rewritten, and we have added four new Figures, namely Figure 6 and 8 in the main article text and Supplementary Figures 4 and 5. Responses to the referee's comments are given below. All changes to the article text are in blue.

Comment 1:

Abstract: "In rare cases, ..." - I would suggest to modify this statement, as quasi-periodic pulsations of the radio emission intensity are a rather frequent phenomenon.

Response:

Agreed, text corrected to 'some cases'.

Comment 2:

p. 2: "... known as a Lotka-Volterra system" - better, "modelled by ..."

Response:

Agreed, text corrected.

Comment 3:

" Solar radio pulsations usually display periodicities of ~0.3-4 seconds" - those pulsations have much longer periods too. I'd suggest to rephrase this statement.

Response:

Agreed, text corrected to more accurately quote the possible ranges of radio pulsations. Now says 'periodicities from seconds to minutes'

Comment 4:

p. 17, the estimation of the sausage mode oscillation period. I would remove the factor of 2.62, as recent works (including the cited in this estimations) demonstrated that this factor appears only in the case of a loop with a step-function perpendicular profile of the fast speed, while for smoother profiles this factor is different. Hence, a more correct estimation would be P less than a/C_F , where C_F is the fast speed. But, this modification does not affect the result and conclusions.

Response:

Agreed, we have altered the calculation of the pulse period and now use a/C_F . This calculation is also significantly more accurate, given the new analysis of the region density and magnetic field,

see Figure 6. The more accurate analysis still support our conclusion that the pulsations was due to an MHD sausage mode oscillation.

Reviewers' comments:

Reviewer #1 (Remarks to the Author):

The authors have the manuscript sufficiently improved.

I recommend publication.

Reviewer #2 (Remarks to the Author):

I thank the authors for being responsive to referee input, which was extensive. While the revision addressed many points raised by the referees, it also raises new questions. In addition, for this referee, the primary objection to the work remains: that the interpretation of the observations is more complex than the data can justify and relies on too many assumptions and guesses to be convincing.

I fully acknowledge that this is an interesting and important observation but, in my opinion, the authors over-interpret the data. I cannot recommend the paper for publication in its present form. I urge them to confine their attention to a clear presentation of the data and a fairly brief and critical assessment of what the data might imply about energy transport, release, and modulation.

Additional comments:

Intro:

p. 2 - "This energy release results in the acceleration of particles to relativistic energies and the generation of high intensity bursts of radio emission." Well, EM radiation across the spectrum including, notably, hard X-ray emission.

p. 2 - "The radio emission is thought to be generated by a plasma instability..." Perhaps at this particular wavelength. At frequencies above 1-2 GHz plasma emission rarely plays any role.

p. 2- "Direct high cadence radio imaging of the behaviour of electron beams during a pulsation event has the ability

to confirm one (or some combination) of these mechanisms." Has the "potential" to confirm one or more mechanism.

General comment: it is puzzling to me that the authors do not make use of RHESSI observations. (They do use FERMI data in their earlier paper on this event.) There is coverage during the time in question. It would be useful to establish whether there is HXR emission in the non-flaring active region or whether the HXR emission in the flaring active region has any connection to the distant pulsating radio source. The same applies to ancillary radio data from, for example, Ondrejov.

Results:

p. 6- "possible cyclotron mechanism or loss-cone instability" The cyclotron mechanism refers to a resonance; a loss-cone provides free energy that could drive either plasma radiation or cyclotron maser radiation.

I don't understand the argument that the type III "most likely" injected energetic electrons into the quiet AR. They type III appears to be directed out into the corona and perhaps the IPM. Is there evidence that it is bidirectional? The rise in radio flux in the quiet AR appears to be well underway by the time the type III occurs.

p. 7&8- The case for reverse drift components - and for downward electron beams - is not very convincing and may be more apparent than real. Were the spectral data corrected for the instrument bandpass?

p. 9- Why do the authors claim that electrons accumulate at the loop footpoint? Downward-moving electrons will either precipitate or mirror back up to greater heights.

Referring to Fig. 5a, the systematic change in the source centroid with an apparent speed of ~ 700 km/s is striking. Why is the motion in all bands confined to the observed direction? How does the detailed timing relate to the EUV wave, which has a speed that is only 310 km/s?

p. 10&11: If the variation in the position of source maxima is apparent (p.10), it is not necessary to quote a speed of $0.1c$ (p.11).

The authors have embraced the presence of electron beams with very little observational evidence. I am also puzzled by the statement: "This shows that while there is pulsing energisation of electrons at P, the electrons accumulate at the adjacent footpoints close to R and Q ...". Energisation due to what?

p. 11- Referring to Fig. 6 the authors claim to identify a magnetic loop in which a sausage mode MHD oscillation could be supported. In panel (a), how does the (systematic) source motion shown in Fig. 5 relate to the NLFFF extrapolation? Is the NLFFF extrapolation consistent with the presence of the loop identified in EUV and shown in Fig. 6c? How do fast electrons gain access to the loop?

p. 13- The authors estimate the density in the loop using a DEM analysis of the AIA data. Is this necessary? They are trying to establish "whether sausage modes could be responsible for the radio pulsations". If the radio emission is due to fundamental plasma radiation the density in the source is $\sim 6 \times 10^8 / \text{cm}^3$, a factor of a few less than they estimate based on the DEM. How does the DEM-based estimate relate to the assumption of a density of 10^9 at the base of the loop?

p. 14- Using an average extrapolated magnetic field strength seems hazardous. Do the authors have any way to determine and propagate uncertainties in estimated quantities here?

Bottom: The authors argue (in the supplementary material) that the pulsations are not well described by an L-V system, yet they comment on p. 9 that the "light-curve of the pulsations shows a sawtooth like pattern ... (this sawtooth characteristic is unusual, as [MHD] pulsations usually have a symmetric rise and decay phase)." This discrepancy between the observations and the MHD interpretation is not adequately addressed.

p. 15- "Observations of such large spectral slopes are rare". Insofar as narrowband emissions are often seen, this statement is not accurate. The "only emission that can explain such spectra..."? The references given are generic and do not really show that ECM amplification of Z-mode is the only possible mechanism.

p. 16- The authors claim that the observed pulsations are highly X-mode polarized. On the other hand, they claim that the reverse-drift emissions are beam-driven fundamental plasma radiation, which should be O-mode polarized. Is such a difference in polarization observed?

p. 17- "roughly coincides" - please be more precise and show, for example, in Fig. 1 of the supplementary material the start time and end time during which the EUV wave could be interacting with the quiet AR.

"10:54 UT"  "12:54 UT"

p. 18&19- the words "most likely" or "likely" are over-used because the various suggestions have not been compellingly demonstrated.

The authors again acknowledge the expectation that an MHD sausage mode would yield a symmetric pulse profile. They then comment that "In this instance given the involvement of electron acceleration and the asymmetric pulse profiles, the modulation may be indirect." What does this mean? How does it yield the observed sawtooth pattern?

The point made above that the ECM source and the electron beam emission should be oppositely polarized is relevant here, too.

Reviewer #3 (Remarks to the Author):

The authors managed to revise the paper according to my comments, and I am happy to recommend the acceptance.

Reviewer #2 (Remarks to the Author):

I thank the authors for being responsive to referee input, which was extensive. While the revision addressed many points raised by the referees, it also raises new questions. In addition, for this referee, the primary objection to the work remains: that the interpretation of the observations is more complex than the data can justify and relies on too many assumptions and guesses to be convincing.

I fully acknowledge that this is an interesting and important observation but, in my opinion, the authors over-interpret the data. I cannot recommend the paper for publication in its present form. I urge them to confine their attention to a clear presentation of the data and a fairly brief and critical assessment of what the data might imply about energy transport, release, and modulation.

Response 1:

We would like to thank the referee for further detailed comments on our revised article. To address the referee's remaining concerns we have re-written parts of the article (indicated in blue text) and included four new Figures, namely a new Figure 8 and Supplementary Figure 5 (originally Figure 5 in main article, but modified and moved to Supplementary Material), as well as new Supplementary Figures 2 and 8. We have also shorted the text and made the discussion section more concise. We are confident that this additional analysis addresses the referee's remaining concerns and is fully supportive of our hypothesis.

Additional comments:

Intro:

p. 2 - "This energy release results in the acceleration of particles to relativistic energies and the generation of high intensity bursts of radio emission." Well, EM radiation across the spectrum including, notably, hard X-ray emission.

Response 2:

Agreed. Text in the introduction has been changed to account for this.

p. 2 - "The radio emission is thought to be generated by a plasma instability..." Perhaps at this particular wavelength. At frequencies above 1-2 GHz plasma emission rarely plays any role.

Response 3:

Agreed. Changed to "The radio emission at metric wavelengths".

p. 2- "Direct high cadence radio imaging of the behaviour of electron beams during a pulsation event has the ability to confirm one (or some combination) of these mechanisms." Has the "potential" to confirm one or more mechanism.

Response 4:

Agreed. Changed to 'potential'.

General comment: it is puzzling to me that the authors do not make use of RHESSI observations. (They do use FERMI data in their earlier paper on this event.) There is coverage during the time in question. It would be useful to establish whether there is HXR emission in the non-flaring active region or whether the HXR emission in the flaring active region has any connection to the distant pulsating radio source. The same applies to ancillary radio data from, for example, Ondrejov.

Response 5:

Agreed. We have now included a new Supplementary Figure 8 showing the pulsations compared to FERMI GBM fluxes at energies 26-50 keV and 50-100 keV. While individual pulsations are not possible to distinguish in the HXR flux (it is primarily noise at such timescales), there is a long time-scale trend in the HXR flux from 12:55-12:58 UT that the radio pulsation flux follows quite closely. Furthermore, we have also included a new analysis of microwave data from RSTN San-Vito, see Supplementary Figure 8. This shows that metric radio emission from the quiescent active region follows closely both HXR and microwave, with the trends being particularly apparent in panel e. Overall, this provides extra evidence in favour of the idea that the pulsation source is due to electrons accelerated to lower altitudes (reverse drift bursts), with the microwave being from gyrosynchrotron emission from the electrons during the precipitation, and the HXR being from the the impact of electrons in the low corona/chromosphere where they emit via the usual thick-target model. During the pulsation time, the microwave and HXR light curve shows no correspondence with the Flare AR radio source (see panel e), so it is unlikely the electrons originate from the flare

itself. As for RHESSI light-curves, it shows much the same behaviour as FERMI GBM, but at lower time resolution so we have not included it here.

The Ondrejov data, along with Orféas, in Figure 1 below shows some sporadic emission from 500 to ~1200 MHz at the time of the pulsations. This may be an indicator of plasma emission due to energetic electrons associated with the pulsations. However, this emission also takes place at the same time as the 'Flare Continuum B' emission from the Flare AR, as described in Carley et al. (2016) for this same event. We cannot specifically associate the Ondrejov bursty emission to energetic electrons accelerated during the pulsations.

Figure 1 — Orféas and Ondrejov RT5 dynamic spectrum during the pulsations. Although there is some sign of bursty plasma emission at higher frequencies than the pulsations, it is not clear that these are related to the pulsations.

Results:

p. 6- "possible cyclotron mechanism or loss-cone instability" The cyclotron mechanism refers to a resonance; a loss-cone provides free energy that could drive either plasma radiation or cyclotron maser radiation.

Response 6:

We have altered the end of this sentence to simply state the emission mechanism may be related to the magnetic field, making no further assumption on the type of resonance or emission mechanism at this point in the article.

I don't understand the argument that the type III "most likely" injected energetic electrons into the quiet AR. They type III appears to be directed out into the corona and perhaps the IPM. Is there evidence that it is bidirectional? The rise in radio flux in the quiet AR appears to be well underway by the time the type III occurs.

Response 7:

Agreed. We have changed the interpretation here to include the slow rise before the type III. The slow rise is an indicator of an increase in activity, while the type III is at least part of this initial activity. Supplementary Figure 8 now shows an analysis of the relationship between metric radio, HXR and microwave. The initial type III labelled as peak 'p1' in panel c is seen in all three, indicating that the electron acceleration is indeed bi-directional, i.e., the microwave and HXR peak indicates electron precipitation and impact in the chromosphere at the same time electrons escape into the corona to produce a type III burst.

As for the rise in radio activity before the type III, this begins at ~12:45 UT and may be related to the rise in activity in the flare AR. For example in Supplementary Figure 8c there is rise in HXR and microwave activity. This is accompanied by a rise in activity both in the Quiet and Flare AR around 12:45 observed in 228 MHz. It is possible that some level of sympathetic flaring took place between these two active regions e.g., due to the eruption and expansion of the Flare AR, the Quiet AR becomes perturbed and increases in activity. In a new Supplementary Figure 2 we plot the integrated flux over the Quiet AR pulsation region from AIA in all EUV channels. There is a slow rise from ~12:45 UT in the hotter filters (131, 94 Angstroms), while the cooler channels decrease in flux — overall this indicates that the emission measure weighted temperature of the plasma is decreasing, however there is some level of activity in the ‘hotter’ channels that indicated a heating.

The rise in EUV flux along with the rise in microwaves, HXR and metric radio is a good indicator that there is some electron acceleration occurring within the Quiet AR in the lead up to the pulsation activity. It might be suggested that electrons accelerated in the flare AR had access to the pulsations site, however there is no correlation between radio activity from the Flare AR and Quiet AR during the pulsations (Supp Figure 8e).

Text has been added to Page 6/7 and the Discussion section of the main article and the Supplementary material to account for these points.

p. 7&8- The case for reverse drift components - and for downward electron beams - is not very convincing and may be more apparent than real. Were the spectral data corrected for the instrument bandpass?

Response 8:

The reverse drift components are weak, but they have the same spectral characteristics as reverse drift bursts, including a drift rate that matches previous observations and a speed (using the density model derived from the DEM estimates) that matches well with the speed predicted for electrons generating such reverse drifts. This is now combined with the close correlation between the HXR and microwave observations as described above, providing more evidence that the pulsations are indeed from continuously accelerated electrons.

Yes, the instrument is calibrated to provide SFUs and the dynamic spectra are background subtracted. This ensures the spectrogram is ‘flattened’ for antenna and instrument response and the features we see are real.

p. 9- Why do the authors claim that electrons accumulate at the loop footpoint? Downward-moving electrons will either precipitate or mirror back up to greater heights.

Response 9:

Agreed. Accumulate is perhaps the wrong word to use. We have changed the wording here from accumulate to ‘concentration’ to indicate that some electrons are located around Q.

Referring to Fig. 5a, the systematic change in the source centroid with an apparent speed of ~700 km/s is striking. Why is the motion in all bands confined to the observed direction? How does the detailed timing relate to the EUV wave, which has a speed that is only 310 km/s?

Response 10:

We have now revised a more detailed analysis of the source motion. Due to these new results we have moved this discussion to the Supplementary Material in the section ‘Apparent motion of the pulsating radio source’. This includes Supplementary Figure 5 that shows all source maxima motion (including large-scale drift eastward and fine timescale back-forth motion during pulsations) is due to flux variation between radio sources at P and Q e.g., the figure shows that a successive flux increase over several minutes at P leads to an apparent source drift eastwards, while the superimposed pulsing of the source at P leads to the back-forth motion evident in Supplementary Movie 1.

We have now removed any mention of actual source motion or speed as this new analysis shows it is merely an apparent drift due to alternating source brightness at Q and P.

SDO AIA_1 335 18-Apr-2014 12:50:02.620 UT

Figure 2 — SDO 131 Angstrom image with NLFFF extrapolated field line shown in grey and the hot loop indicated. The green and orange contours show the positive and negative polarities of the magnetogram, respectively.

p. 10&11: If the variation in the position of source maxima is apparent (p.10), it is not necessary to quote a speed of 0.1c (p.11).

Response 11:

Agreed. As mention in previous comment, and now shown in Supplementary Figure 5, all source motion is apparent, so we have removed any mention of source speed to avoid confusion.

The authors have embraced the presence of electron beams with very little observational evidence. I am also puzzled by the statement: "This shows that while there is pulsing energisation of electrons at P, the electrons accumulate at the adjacent footpoints close to R and Q ...". Energisation due to what?

Response 12:

Given the new evidence of the pulsations overall trend being well correlated with HXR and microwave data (over long-timescales), we now have a stronger indication that the activity involved electron acceleration. This combined with the fact we also see reverse drift bursts is good evidence to suggest the presence of electron beams.

As for the energisation, it is unclear as to what specifically accelerates these electrons. We have been careful not to over-interpret the data and merely mention that because we observe evidence for electron beams at a magnetic null-point point in the corona, magnetic reconnection is a viable mechanism. For clarity, a new Figure 8 in the main article now indicates the proposed mechanism, with positions of oscillating loop, site of reconnection, and P and Q indicated.

p. 11- Referring to Fig. 6 the authors claim to identify a magnetic loop in which a sausage mode MHD oscillation could be supported. In panel (a), how does the (systematic) source motion shown in Fig. 5 relate to the NLFFF extrapolation? Is the NLFFF extrapolation consistent with the presence of the loop identified in EUV and shown in Fig. 6c? How do fast electrons gain access to the loop?

Response 13:

As mentioned above and now shown in the new Supplementary Figure 5, the systematic motion is an apparent one. There is no observable overall motion of the site of energetic electrons towards the east.

It is difficult to resolve loops on such size-scales in the NLFFF extrapolation. Despite this loop not being resolved specifically, it is present in the EUV data and clearly has magnetic footpoints in the magnetogram, shown in Figure 2 above. We have taken an average of the magnetic field strength in a box that approximates the size of this loop (see Response 15 below for interpretation of uncertainties involved in this extrapolation).

From Figure 8 of the main article, access to the surrounding environment (footpoints at Q or the loop itself) could be due to reconnection at the null point, with field lines around Q or the loop being involved in this reconnection. Again we stress that reconnection is not specifically observed, being mentioned only in the discussion section of the paper as a possibility. Lack of observation of reconnection however does not change the overall results of the paper.

p. 13- The authors estimate the density in the loop using a DEM analysis of the AIA data. Is this necessary? They are trying to establish "whether sausage modes could be responsible for the radio pulsations". If the radio emission is due to fundamental plasma radiation the density in the source is $\sim 6 \times 10^8 / \text{cm}^3$, a factor of a few less than they estimate based on the DEM. How does the DEM-based estimate relate to the assumption of a density of 10^9 at the base of the loop?

Response 14:

The density estimate was necessary to derive the Alfvén speed, which then gives an estimate of the sausage mode period that such a loop would generate.

The base density estimate was based on the order of magnitude of the loop density e.g., we assume that the base density of the corona surrounding/outside the loop region is of a similar order of magnitude to the small loop. From this base density we estimate that $6 \times 10^8 \text{ cm}^{-3}$ is at $\sim 15 \text{ Mm}$ altitude (slightly above the loop). Using a base density of $2 \times 10^9 \text{ cm}^{-3}$ (small loop average density) the source would be placed at 37 Mm , still within the vicinity of the loop. Considering the immediate environment of the loop has no greater density than the loop itself (the loop represents a slightly over-dense structure in this environment) this height represents a maximum altitude of 228 MHz based on a hydrostatic equilibrium model.

p. 14- Using an average extrapolated magnetic field strength seems hazardous. Do the authors have any way to determine and propagate uncertainties in estimated quantities here?

Response 15:

While a range of estimates for the loop properties is possible given the range of parameters of width, temperature, density and magnetic field, a true estimate of uncertainty is not possible due to the lack of uncertainty analysis from the NLFFF magnetic field values (Wiegmann et al. 2012). That said, given the relationship of period P to the Alfvén speed v , sound speed c and loop width a , a propagation of uncertainty shows that the P fractional uncertainty is given by

$$\frac{\Delta P}{P} = \sqrt{\left(\frac{\Delta a}{a}\right)^2 + \left(\frac{v \Delta v}{v^2 + c^2}\right)^2 + \left(\frac{c \Delta c}{v^2 + c^2}\right)^2}$$

Now, given the relationship of Alfvén speed v , magnetic field B and density n , and the sound speed c dependency on temperature T , the fractional uncertainties of these quantities are

$$\frac{\Delta v}{v} = \sqrt{\left(\frac{\Delta B}{B}\right)^2 + \left(\frac{\Delta n}{2n}\right)^2} \quad \frac{\Delta c}{c} = \sqrt{\left(\frac{\Delta T}{2T}\right)^2}$$

With $\Delta n/n=0.07$, $\Delta T/T=0.2$ and $\Delta a/a=0.1$, the relationship between the P and B fractional uncertainty is shown in Figure 3 below. This shows that $\Delta P/P \sim \Delta B/B$, hence P is not highly sensitive to the B uncertainty. We have included this in the Methods section of the paper.

Figure 3 — Relationship between fractional uncertainties of sausage mode period and magnetic field strength. For the quoted uncertainties of density, temperature and loop width, $\delta_P/P \sim \delta_B/B$.

Bottom: The authors argue (in the supplementary material) that the pulsations are not well described by an L-V system, yet they comment on p. 9 that the "light-curve of the pulsations shows a sawtooth like pattern ... (this sawtooth characteristic is unusual, as [MHD] pulsations usually have a symmetric rise and decay phase)." This discrepancy between the observations and the MHD interpretation is not adequately addressed.

Response 16:

In the Discussions & Conclusions we gave a reason for the pulse asymmetry and the relation to an MHD wave mechanism. This involved the MHD wave being only involved in the electron acceleration and indirectly involved in radio emission modulation. We have clarified the text in the Discussion & Conclusion section to make this point clearer.

p. 15- "Observations of such large spectral slopes are rare". Insofar as narrowband emissions are often seen, this statement is not accurate. The "only emission that can explain such spectra..."? The references given are generic and do not really show that ECM amplification of Z-mode is the only possible mechanism.

Response 17:

Narrow band emissions are often seen but to our knowledge the measurement of spectral slopes of 21 and -13 are rare in the literature. We have removed "the only emission that can explain...". We now say that such strong spectral slopes can be explained theoretically using plasma emission from a loss-cone instability (Z-mode). This theoretical explanation is supplemented by an ensemble of other pieces of observational evidence (source position at magnetic footpoints, association with B-field, polarisation, and ratio of plasma to cyclotron frequency) allowing us to conclude that this emission is due to the Z-mode of loss-cone instability.

p. 16- The authors claim that the observed pulsations are highly X-mode polarized. On the other hand, they claim that the reverse-drift emissions are beam-driven fundamental plasma radiation, which should be O-mode polarized. Is such a difference in polarization observed?

Response 18:

Indeed fundamental plasma from type III radio burst is expected to be polarised in the sense of the O-mode. In this case, any emission associated with the pulsations is in X-mode. As explained in Melrose et al. (1978) and Dulk & Suzuki (1980) the polarisation of harmonic plasma emission can

be O or X-mode depending on viewing angle. For viewing angles >90 degrees, the sense of polarisation can reverse, meaning harmonic plasma emission can be in the sense of the X-mode. Since these are reverse drift bursts and assumed to be downward propagating electrons on a negative magnetic field, the viewing angle for the emission is expected to be >90 degrees and in the opposite sense to what we would expect from a type III. Hence it is theoretically possible for the emission to be plasma emission and X-mode in the case reverse drift bursts.

p. 17- "roughly coincides" - please be more precise and show, for example, in Fig. 1 of the supplementary material the start time and end time during which the EUV wave could be interacting with the quiet AR.

Response 19:

The approximation here was due to the fact the EUV wave is faint and its front is not very well defined in space. From Figure 7 of the main article, the EUV front makes contact with the quiet AR (~100 Mm) quite early in the event, around 12:45 UT. This could explain the rise in emission at this time observed in radio and HXR in Supplementary Figure 8 and in EUV in Supplementary Figure 2. It first interacts with the pulsation region at ~12:54 UT, when the pulsations begin. One of the last interactions the wave has with the Quiet AR is at 13:03 UT, seen in Supplementary Figure 1c, after which the radio emission diminishes. We have included these points in new text in the 'EUV wave kinematics' section.

"10:54 UT"  "12:54 UT"

Response 20:

Fixed typo.

p. 18&19- the words "most likely" or "likely" are over-used because the various suggestions have not been compellingly demonstrated.

Response 21:

Pages 18 & 19 are reserved for a discussion section of the paper where we postulate the most likely interpretation of the observations and analysis we have presented. We have been careful here not to make definitive statements. For example, we have shown that there is good evidence to suggest that an MHD wave was involved in these pulsations, and there is evidence to the contrary for the LV system. Use of the word 'likely' in this instance means there is more supportive evidence for the MHD oscillation mechanism. We have changed the language on page 18 & 19 in this regard

The authors again acknowledge the expectation that an MHD sausage mode would yield a symmetric pulse profile. They then comment that "In this instance given the involvement of electron acceleration and the asymmetric pulse profiles, the modulation may be indirect." What does this mean? How does it yield the observed sawtooth pattern?

Response 22:

We have now added further detail to the Discussion & Conclusion section and a new Figure 8 to clarify this point. We also mentioned on page 19 the sawtooth rise-phase being due to build up of loss-cone emission (and sometimes direct observation of the reverse drift bursts), while the sawtooth decay phase is due to loss-cone emission quenching via particle injection. This section also makes reference to the Supplementary Material where we discuss the theoretical details and plausibility of this scenario in the 'Viability of the sudden reduction mechanism' section.

The point made above that the ECM source and the electron beam emission should be oppositely polarized is relevant here, too.

Response 23:

As in the Response 18, for reverse drift bursts the sense of polarisation for plasma emission can be X-mode in the case of reverse drift bursts.

Reviewers' comments:

Reviewer #2 (Remarks to the Author):

I am afraid the authors and I are not really converging. Below are a few additional comments on the various components of their scenario. If I've mischaracterized or misunderstood aspects of the scenario after three readings, it is indicative of how hard it might be for a typical reader to follow. I leave it to the editor on how they wish to proceed.

The authors suggest the following scenario to account for the observed radio imaging (NRH) and spectroscopic (Orféés) observations:

- An MHD sausage mode oscillation is excited in a coronal loop by the passage of an EUV wave driven out from a large flare in an adjacent active region
 - o I think the authors have fairly convincingly shown that the passage of the EUV wave is associated with the onset of the pulsations
- The oscillation leads to the periodic acceleration of electrons near a magnetic null point where intense 228 MHz emission is produced
 - o The weak pulsations in source Q could just be a sidelobe response to the strong pulsations in P (p. 7)
 - o In Fig. 5, panels a-c, where are P and Q relative to the candidate pulsating loop?
 - o The authors claim that the intense, pulsating 228 MHz emission is fundamental plasma radiation (p. 12). What produces (and confines) the underlying spectrum of plasma waves? What is the degree and sense of circular polarization of the pulsations? Same as it is for the emission near 210 MHz (Fig. 6)?
- The accelerated electrons stream down from P toward the foot points of adjacent loops at site Q; the downward streaming electrons produce reverse-drift type III radio emission
 - o Are the reverse-drift type IIIs fundamental plasma radiation? What is their polarization degree and sense?
 - o Is there any evidence for classical type IIIs drifting from 228 MHz to lower frequencies in the spectrum? What is the emission <200 MHz in Fig. 7 in the Supp. Materials?
 - o Looking at the Fig. 4, lower panel, it appears that there is a radio signature correlated with the type IIIs at frequencies up to 327 MHz. Can't the 327 MHz data be filtered (remove the

background with a running boxcar or similar) in order to isolate the location, timing, and polarization of the purported reverse-drift type IIIs at 327 MHz source relative to P and Q?

o I asked the authors why they hadn't used FERMI and RHESSI HXR observations. They now make use of HXR and microwave light curve data and claim that it is well correlated with the observed meter-wavelength phenomena. But what about RHESSI maps? It would be extremely interesting and important to see where an associated HXR source is located relative to P and Q.

• At site Q, a loss cone is set up that is unstable to the production of Z-mode waves that are subsequently converted to electromagnetic waves – the emission is imaged at 228, 298, and 327 MHz

o I find the section beginning on p. 13 and continuing (emission mechanism and loss-cone instability modulation) to be very confusing. The claim is that a "secondary mechanism" must be in play to account for the strong narrow-band emission that peaks at ~210 MHz and the authors suggest a loss-cone instability. Isn't that source P? What sets up a loss cone there?

o Bottom of p. 14: the ratio of the electron plasma frequency to the electron gyrofrequency is given as 1.6. The statement is made that the ratio is 2 for harmonic emission. Not 3.2? Harmonic plasma radiation is near 2 times the plasma frequency.

o The authors suggest that z-mode waves are amplified by the loss-cone and then coalesce, producing x-mode harmonic radiation. They also point out the fact that the 228-327 MHz emission seems to be associated with loop footpoints at source Q. Are the authors claiming loss-cone emission at both points P and Q and that it is harmonic plasma radiation? It is also hard to see how this is consistent with statements about the 228 GHz source on p. 12 since the harmonic emission at source Q would have a lower density than the fundamental emission at source P.

o It is hard to see the relevance of several of the references cited in support of the above claims (e.g., the regimes considered by the references relevant to polarization [51-53] don't seem to be relevant to conditions in the source.

• The loss cone is periodically quenched by the periodic introduction of fresh electrons streaming down from the pulsating acceleration site P

o It seems the authors argue that the loss-cone-driven emission is relatively broadband (100 MHz) in Q. Either the source is extended (plasma radiation from a range of sites) or it is intrinsically broadband, perhaps due to conditions described by Winglee and Dulk (1986).

o Why is the emission <200 MHz also subject to sudden reductions?

Reviewers' comments:

Reviewer #2 (Remarks to the Author):

I am afraid the authors and I are not really converging. Below are a few additional comments on the various components of their scenario. If I've mischaracterized or misunderstood aspects of the scenario after three readings, it is indicative of how hard it might be for a typical reader to follow. I leave it to the editor on how they wish to proceed.

Response 1:

We would once again like to thank the referee for their time and effort in reviewing our article. We have rewritten parts of the article (text in blue) to make it as concise and clear as possible for the general reader. In particular we have simplified the 'Emission mechanism...' and 'Discussion...' sections by moving some of the finer details into the Supplementary Material.

The authors suggest the following scenario to account for the observed radio imaging (NRH) and spectroscopic (Orféés) observations:

- An MHD sausage mode oscillation is excited in a coronal loop by the passage of an EUV wave driven out from a large flare in an adjacent active region*
 - o I think the authors have fairly convincingly shown that the passage of the EUV wave is associated with the onset of the pulsations*
- The oscillation leads to the periodic acceleration of electrons near a magnetic null point where intense 228 MHz emission is produced*
 - o The weak pulsations in source Q could just be a sidelobe response to the strong pulsations in P (p. 7)*

Response 2:

For several reasons, the source at Q cannot be a sidelobe of P. Firstly, Q exists before P is present. Sidelobe sources cannot exist without the source of the primary beam. Secondly, side lobe source flux variation tends to be in-phase with the primary source. There are times during the pulsations when Q is directly out of phase with P e.g., one source often exists without the other. This means they are two independent, real sources.

o In Fig. 5, panels a-c, where are P and Q relative to the candidate pulsating loop?

Response 3:

We have indicated where P and Q are on panel a. This makes it clear that the proposed pulsating loop is directly in the vicinity of P.

o The authors claim that the intense, pulsating 228 MHz emission is fundamental plasma radiation (p. 12). What produces (and confines) the underlying spectrum of plasma waves? What is the degree and sense of circular polarization of the pulsations? Same as it is for the emission near 210 MHz (Fig. 6)?

Response 4:

For reverse drift bursts, the production of plasma emission is via the bump-on-tail instability, followed by induced scattering of Langmuir waves e.g, the standard plasma emission mechanism. As for the loss-cone emission, energetic electrons with an initial power law distribution in trapped coronal loops will eventually develop into a loss-cone distribution that becomes unstable to the production of Langmuir waves. The instability can last on the order of minutes (Benz & Tarnstrom 1976) and result in the production of radio emission via Langmuir wave scattering. In this sense, energetic electrons initially trapped in coronal loops produce the loss-cone emission. The fresh/repeated injection of new electrons would temporarily quench this loss cone.

Benz & Tarnstrom (1976) also discuss that the loss cone instability re-establishes itself once the injected electrons have escaped the trap (if they have a fast enough $v_{||}$). Some of the newly injected electrons may also mirror (those with a smaller $v_{||}$) and add to the original loss-cone distribution which was already established. This is another means by which the original loss cone distribution could be sustained on the order of minutes

From Supplementary Figure 4, the pulsations at 228MHz are negative Stokes V of up to 60-70%. We cannot observe sources at 210 MHz (outside any NRH observing bands), but it is likely they are the same as 228 MHz e.g., the 228 MHz NRH source and the 210 MHz Orféés light curve are identical, so they are from the same source. Unfortunately Orféés are currently uncalibrated in Stokes V for this event.

- *The accelerated electrons stream down from P toward the foot points of adjacent loops at site Q; the downward streaming electrons produce reverse-drift type III radio emission*
o Are the reverse-drift type IIIs fundamental plasma radiation? What is their polarization degree and sense?

Response 5:

It is not possible to definitively say whether they are fundamental or harmonic. A common assumption for plasma radiation is that it is harmonic, given that the emission has escaped the corona. We observe the reverse drifters at the time of the pulsations, which in NRH images are negative Stokes -V polarised. It is possible that the reverse drift bursts are themselves polarized, but given the fact we cannot observe them at all frequencies in their bandwidth using NRH, we cannot say for sure what their degree of polarisation is. Melrose et al. (1978) and Willes & Melrose (1996) describe that emission from the coalescence of Langmuir waves will be X-mode polarised if the Langmuir waves are isotropic. Since the reverse drifters propagate away from the observer, one would expect their associated Langmuir waves to be isotropic (because we observe the emission), and hence X-mode. This assertion applies only to harmonic emission.

- o Is there any evidence for classical type IIIs drifting from 228 MHz to lower frequencies in the spectrum? What is the emission <200 MHz in Fig. 7 in the Supp. Materials?*

Response 6:

There is evidence in the form of the initial type III at 12:56:10 UT, which can be seen in Supplementary Figure 8 in the Orféas and NDA dynamic spectrum, which has a corresponding signature in the HXR and microwave data. However, there is no evidence of forward drift bursts related to the reverse drifters. The emission <200 MHz may be part of the entire continuum of emission produced during the ECM Z-mode.

- o Looking at the Fig. 4, lower panel, it appears that there is a radio signature correlated with the type IIIs at frequencies up to 327 MHz. Can't the 327 MHz data be filtered (remove the background with a running boxcar or similar) in order to isolate the location, timing, and polarization of the purported reverse-drift type IIIs at 327 MHz source relative to P and Q?*

Response 7:

Indeed part of the 327 MHz emission in panel d may be from the reverse drifters. It is extremely weak in Orféas and difficult to isolate. The spectrum as it exists has been background subtracted, smoothed and intensity scaled. After much experimentation this is the best enhancement of the dynamic spectrum we can achieve to observe the reverse drifters. Due to the weakness of these bursts at 327 MHz it is difficult to isolate them in time and compare them to the source in the images. This is made more difficult by the fact that the 327 MHz source in the images does not show pronounced pulsations, so establishing a temporal relationship with the reverse drifters may not be possible. That said, 327 MHz is also negative Stokes V polarised, so is related to whichever mechanism is responsible for the lower 228 MHz pulsating source (Z-mode ECM).

- o I asked the authors why they hadn't used FERMI and RHESSI HXR observations. They now make use of HXR and microwave light curve data and claim that it is well correlated with the observed meter-wavelength phenomena. But what about RHESSI maps? It would be extremely interesting and important to see where an associated HXR source is located relative to P and Q.*

Response 8:

RHESSI is an indirect imager and has a limited dynamic range for imaging. Moreover, apart from snapshot imaging (requiring a high count-rate) "regular" imaging cannot be achieved at a time resolution less than 4s. Furthermore, at the time of the radio pulsations (sources P and Q) strong X-ray emission (as well as radio) is still observed in the "flaring" active region. In these conditions RHESSI maps cannot show X-ray emissions associated with P and Q, unless they are stronger than the emissions coming from the flaring AR.

- *At site Q, a loss cone is set up that is unstable to the production of Z-mode waves that are subsequently converted to electromagnetic waves – the emission is imaged at 228, 298, and 327 MHz*

o I find the section beginning on p. 13 and continuing (emission mechanism and loss-cone instability modulation) to be very confusing. The claim is that a "secondary mechanism" must be in play to account for the strong narrow-band emission that peaks at ~210 MHz and the authors suggest a loss-cone instability. Isn't that source P? What sets up a loss cone there?

Response 9:

We have rewritten and simplified this section so it is easier to follow. Many of the details about dominant ECM modes and ratio of plasma to gyro frequencies have been moved into the Supplementary Material.

Indeed the loss-cone emission can exist anywhere there is opportunity for energetic electrons to mirror in the vicinity of P and Q e.g., at any magnetic footpoints in this region. Due to the beam size of NRH, it is not possible to say precisely where this is, but given the number of footpoints in the vicinity of P and Q from the magnetic field extrapolations, there is ample opportunity for electrons to encounter converging magnetic fields and mirror. This is evidenced by the pulsing 228 MHz emission in the region of both P and Q. We have rewritten parts of the text to state that the loss-cone emission (at 228 MHz) occurs between P and Q, not

strictly one site or the other. The modulation of this emission occurs by electron acceleration at P and the injection of electrons onto field lines in the vicinity of P and Q that host loss-cone emission.

o Bottom of p. 14: the ratio of the electron plasma frequency to the electron gyrofrequency is given as 1.6. The statement is made that the ratio is 2 for harmonic emission. Not 3.2? Harmonic plasma radiation is near 2 times the plasma frequency.

Response 10:

This ratio is not a simple double because the height of the harmonic emission changes. This makes the gyrofrequency and hence the ratio change. We have changed the text to make this clearer. Note these details have largely been moved to the Supplementary Material and expanded upon.

o The authors suggest that z-mode waves are amplified by the loss-cone and then coalesce, producing x-mode harmonic radiation. They also point out the fact that the 228-327 MHz emission seems to be associated with loop footpoints at source Q. Are the authors claiming loss-cone emission at both points P and Q and that it is harmonic plasma radiation?

Response 11:

Yes, we claim that harmonic emission from ECM in the vicinity of both P and Q, due to there being many regions where such an emission mechanism can take place. We have clarified points in the text to say that loss-cone emission can take place around P and Q. That said, some of the emission at Q (especially at frequencies >298 MHz) is unaffected by electron injection and does not show a pulsation behaviour. Figure 8 of the main article has been slightly modified to take into account this point.

It is also hard to see how this is consistent with statements about the 228 GHz source on p. 12 since the harmonic emission at source Q would have a lower density than the fundamental emission at source P.

Response 12:

If the emission at P and Q is plasma emission at the same harmonic, then emission at Q may be at a similar height for the same frequency. As explained above, because the emission is X-mode polarised it may be at the harmonic.

o It is hard to see the relevance of several of the references cited in support of the above claims (e.g., the regimes considered by the references relevant to polarization [51-53] don't seem to be relevant to conditions in the source.

Response 13:

Melrose et al. (1978) state that is only within relatively narrow viewing angles to the magnetic field (<20 degrees) that one would expect polarisation to be O-mode for second harmonic plasma emission. Outside of this it can be X-mode, especially if the Langmuir wave distribution that causes the emission is isotropic (Willes & Melrose 1996).

Secondly, for the loss-cone emission in particular, the physical conditions described in Melrose et al. (1978) are relevant to this study. For example, in their Figure 2, they consider isotropic distributions of Langmuir waves from a loss-cone (top panel). With viewing angles greater than 70 degrees to the magnetic field and loss cone angles of 30 degrees, the value for 'a' is in the region -1.0. Now assuming harmonic emission the ratio gyro/plasma frequency is 0.5 from our study, meaning the polarisation will be -0.5 (X-mode), which is close to the values of polarisation we observe. Furthermore, the authors discuss the expectation of X-mode polarisation of loss-cone emission from type IVs in the last paragraph of their paper.

• The loss cone is periodically quenched by the periodic introduction of fresh electrons streaming down from the pulsating acceleration site P

o It seems the authors argue that the loss-cone-driven emission is relatively broadband (100 MHz) in Q. Either the source is extended (plasma radiation from a range of sites) or it is intrinsically broadband, perhaps due to conditions described by Winglee and Dulk (1986).

o Why is the emission <200 MHz also subject to sudden reductions?

Response 14:

Yes there may be plasma emission from a range of heights that results in the emission from 200-327 MHz. Given the electrons propagated onto different field lines in the vicinity of P and Q, they each may encounter slightly different densities. This would result in the emission occurring over a range of frequencies. This idea was first suggested in the original Benz & Tarnstrom (1976) paper. While it remains a possibility, we also cannot rule out a broadband source due to the intrinsic nature of the emission (Winglee and Dulk 1986), as stated by the referee. However, lack of clarity between these two points does not change the general result of the paper. We have added text to the Discussion section and slightly changed the schematic in Figure 8 to show that some of the emission at Q comes from deeper in the corona and to the east of the electron injection point, hence it is unaffected by the electron injection. This would explain the less pronounced

pulsations around Q at the higher frequencies. Finally, if the emission at frequencies below 200 MHz is also from the Z-mode process, then interruptions to this process will also interrupt his emission.